Article 

# *hapln1a*$^+$ cells guide coronary growth during heart morphogenesis and regeneration

Jisheng Sun ◉[1], Elizabeth A. Peterson ◉[1], Xin Chen ◉[1] & Jinhu Wang ◉[1]✉

Although several tissues and chemokines orchestrate coronary formation, the guidance cues for coronary growth remain unclear. Here, we profile the juvenile zebrafish epicardium during coronary vascularization and identify *hapln1a*$^+$ cells enriched with vascular-regulating genes. *hapln1a*$^+$ cells not only envelop vessels but also form linear structures ahead of coronary sprouts. Live-imaging demonstrates that coronary growth occurs along these pre-formed structures, with depletion of *hapln1a*$^+$ cells blocking this growth. *hapln1a*$^+$ cells also pre-lead coronary sprouts during regeneration and *hapln1a*$^+$ cell loss inhibits revascularization. Further, we identify *serpine1* expression in *hapln1a*$^+$ cells adjacent to coronary sprouts, and *serpine1* inhibition blocks vascularization and revascularization. Moreover, we observe the *hapln1a* substrate, hyaluronan, forming linear structures along and preceding coronary vessels. Depletion of *hapln1a*$^+$ cells or *serpine1* activity inhibition disrupts hyaluronan structure. Our studies reveal that *hapln1a*$^+$ cells and *serpine1* are required for coronary production by establishing a microenvironment to facilitate guided coronary growth.

Adult humans lack the capacity to regenerate lost contractile tissues and this deficiency leads to heart failure, which has no final therapeutic strategy available and remains the leading cause of death worldwide[1,2]. Recent studies demonstrated mammalian cardiomyocyte turnover occurs throughout life[3–5], suggesting an endogenous regeneration program exists in mammals. However, this program cannot be efficiently initiated or completed to effectively restore lost cardiac tissue. Elucidating the mechanisms of endogenous cardiac regeneration will facilitate the development of regenerative therapeutic strategies. Previous studies in neonatal mice, zebrafish, newts, and urodele amphibians demonstrated their robust capacity to significantly regenerate damaged cardiac tissues[6–9]. These reports also identified multiple cell types and molecules with roles in endogenous heart regeneration, but the details of their coordination to ensure successful regeneration is largely unknown.

As a crucial step in mammalian heart repair and regeneration, coronary revascularization therapies are an effective means to treat ischemic heart disease and are considered a promising strategy for myocardial restoration[10–12]. Currently, only large-caliber vessels can be targeted in therapy and the mechanisms of coronary revascularization

during regeneration are still under investigation. Fully characterizing coronary growth will help in the pursuit of better tools and interventions to stimulate efficient heart repair and regeneration. Cardiac tissues and macrophages were previously identified as essential factors for coronary expansion, migration, patterning, and remodeling[13–16]. Mitogens and chemokines like *VEGF*, *IGF*, and *Cxcl12* play vital roles during coronary growth, and some of these factors are speculated to guide coronary expansion[12,15–17]. As these proposed guiding molecules are not well illustrated in their derived tissues, the direct interaction of these cues with coronary growth tips has yet to be observed.

The epicardium is a mesothelial cell layer enveloping the outer surface of the vertebrate heart and conducts multiple functions in cardiac development and regeneration[18–21]. Due to their functional roles, there is increasing interest in targeting epicardial cells to promote heart repair[22]. However, the epicardium itself is a heterogeneous tissue and which subpopulations offer benefits to cardiac regeneration events is far from clear. Recent gene expression profiling analyses identified an epicardial cell subset that surrounds and supports proliferating cardiomyocytes during myocardial compaction and regeneration[23], suggesting a "niche" effect of an epicardial

[1]Cardiology Division, School of Medicine, Emory University, Atlanta, GA 30322, USA. ✉e-mail: jinhu.wang@emory.edu

subpopulation on generating new contractile tissues. As the epicardium is also required for coronary vascularization and revascularization[12,24], there is likely an epicardial cell cluster responsible for coronary growth during heart morphogenesis and regeneration. Presently, an epicardial cell subtype that supports vascular growth has yet to be identified.

To explore the effect of epicardial cell clusters on coronary production, we initially focused on coronary vascularization as tissue injury promotes organ developmental programs to restart. Moreover, coronary vascularization encompasses a simpler environment than revascularization, which incurs strong injury-induced stress/inflammatory responses. Here, we first examined epicardial cells with scRNA-seq analysis during the juvenile stage when robust coronary vascularization occurs in compact muscles and detected *hapln1a*+ epicardial cells enriched with angiogenesis genes. Then, fluorescence reporter assays indicated that *hapln1a*+ cells not only surround coronary vessels but also form pre-leading linear structures in advance of coronary sprouts. Next, we visualized coronary growth in juvenile heart surfaces and observed coronary extension along *hapln1a*+ cellular structures, while *hapln1a*+ cell depletion blocked such coronary growth. We also discovered *hapln1a*+ cells exist ahead of coronary sprouts in the regenerating area of adult hearts and depleting *hapln1a*+ cells blocked coronary revascularization. After examining gene expression in *hapln1a*+ cells from juvenile and regenerating hearts with scRNA-seq and in situ hybridization analyses, we detected *hapln1a*+ cells locally expressed *serpine1*, a gene previously implicated in cancer angiogenesis, metastasis, and with expression around coronary sprouts. Pharmacological inhibition of *serpine1* function resulted in coronary growth blockage during heart morphogenesis and regeneration. Furthermore, we found that the *hapln1a* substrate, hyaluronan (HA), formed a linear structure along and preceding the coronary vessels. Depletion of *hapln1a*+ cells or inhibition of *serpine1* activity disrupted the HA structure. Our results demonstrate that *hapln1a*+ cells are required for coronary production by creating a local environment to facilitate coronary growth and reveal a key cellular player of coronary regrowth during heart regeneration.

## Results

### *hapln1a*+ epicardial cell clusters in juvenile hearts are enriched with angiogenesis factors

Zebrafish coronary growth establishes a dense vasculature network from 5 to 6 weeks post-fertilization (wpf) until the adult stage, a period during which latent epicardial clusters are also undergoing development[23]. As cardiac regeneration recapitulates many aspects of heart morphogenesis and detection of epicardial cell clusters at the juvenile stage has not been reported, we performed scRNA-seq analysis with *tcf21*+ cells[25] in zebrafish hearts at 7 wpf (Supplementary Fig. 1) to explore juvenile epicardial clusters and their potential effects on coronary growth. After performing stringent filtering to discard a small number of non-epicardial cell types like cardiomyocytes (*myl7*)[26], endothelial cells (*fli1a*)[16], and hematopoietic cells (*gata1a, lcp1*)[27,28], we obtained high-quality transcriptomes of 733 *tcf21*+ cells. Unsupervised clustering identified 6 clusters, with each cluster possessing characteristic gene expression patterns (Fig. 1a–c and Supplementary Fig. 2). Clusters 1 and 3 revealed a high degree of expression overlap for genes associated with vascular growth and extracellular matrix (ECM) organization, as both clusters expressed high levels of *fn1a* and *ntn1*[29,30]. Although it is unclear whether they represent distinct cellular identities, we observed a notable difference in cluster 3 with high levels of expression of genes related to cell proliferation, such as *frzb*, *mustn1a*, and *tgfbi*[31–33]. Cluster 2 cells predominantly displayed retinoic acid metabolic processing genes, such as *aldh1a2* and *crabp1a*[34,35]. Cluster 4 cells highly expressed several genes implicated in translation, such as *rpl39* and *rps21*[36,37]. The expression of *ciarta* and *dbpβ* in cluster 5 suggests an involvement in the circadian rhythm process[38,39]. Cluster

6 cells expressed *cxcl12b* and *sema4ba*, indicative of chemotaxis roles[40,41]. Among these cells, clusters 1, 3, and 6 displayed enrichment in vascular growth-related genes (Fig. 1c). As cluster 1 is the most heavily represented of these 3 cell states, we first focused on cluster 1 and observed high expression of *hapln1a* in this cluster[23], with slight expression in clusters 3, 4, and 5 (Fig. 1d and Supplementary Fig. 3). To determine whether *hapln1a*+ cells have elevated expression of angiogenesis-related genes, we compared gene expression in *tcf21*+/ *hapln1a*+ cells vs. *tcf21*+/*hapln1a*- cells (Fig. 1e). These results showed an enrichment of angiogenesis-related genes such as *angptl2b*, *dcn*, *cxcl12a*, *fstl1b*, and *rspo3*[42–46] in *hapln1a*+ cells, and gene ontology analysis further revealed angiogenesis as a top enrichment factor in *hapln1a*+ cells (Fig. 1e and f), suggesting *hapln1a*+ cells regulate coronary vascularization.

### *hapln1a*+ cells pre-lead coronary sprouts during vascularization

To determine the interaction between *hapln1a*+ cells and coronary vessels, we examined juvenile *hapln1a:mCherry;deltaC:EGFP* fish, whose transgenes have been utilized to visualize *hapln1a*+ cells and coronary endothelial cells, respectively[23,47]. We observed EGFP fluorescence signals surrounded by mCherry+ signals (Fig. 2a), indicating that *hapln1a*+ cells wrap around coronary vessels and behave as perivascular cells[48]. We then analyzed the scRNA-seq data for perivascular cell markers and observed *pdgfrβ* expression in clusters 1 and 6[49] (Supplementary Fig. 4). Interestingly, we further noticed that mCherry+ cells formed linear structures located ahead of EGFP+ cells (Fig. 2b). These observations were surprising, as perivascular cells are known for their recruitment to newly formed vessels for stabilization and maturation, and they normally develop after or in parallel with vessel sprouts[50,51]. To confirm these observations and determine whether we uncovered a rare phenomenon, we categorized and quantified three *hapln1a*+ cell position types according to their localization with coronary sprouts in juvenile ventricles: *hapln1a*+ cells trailing, in parallel, or in advance of *deltaC*+ cells (Fig. 2c). The results indicated that the position type of *hapln1a*+ cells in advance of *deltaC*+ cells is dominant (82%) (Fig. 2d), suggesting a spatial correlation of *hapln1a*+ cells with coronary sprouts during vascularization.

### Coronary sprouts are not tightly correlated with macrophages, nerves, or the myocardium

As previous studies suggest macrophages, nerves, and cardiomyocytes affect coronary vasculature formation[13,14,17,52–54], we asked whether these tissues have a spatial correlation with coronary sprouts. To assess the association of coronary growth with macrophages, we extracted juvenile *deltaC:EGFP* fish hearts and performed IB4 staining for macrophage visualization[55]. We observed some macrophages close to coronary vessels (Fig. 2e), but we seldomly detected macrophages located directly ahead of coronary sprouts.

Due to the observed alignment of blood vessels and nerves in many adult peripheral tissues, it is speculated that these cells follow each other's growing path[56]. As zebrafish ventricular innervation occurs during coronary vascularization[57], we then asked if coronary growth associates with nerve fibers. To test this possibility, we isolated hearts from juvenile *deltaC:EGFP* fish and performed antibody staining against acetylated α-tubulin (AcT) and human neuronal C/D (Hu) to visualize heart innervation[57,58], an established label combination which has previously been used to detect neuronal somas in neuroanatomical studies in the zebrafish intestine and goldfish heart[59–61]. We observed some EGFP+ signals and AcT-Hu+ cells in parallel in the ventricular surface, but EGFP+ cell extensions were not close to AcT-Hu+ cells (Fig. 2f). These observations indicate alignment of zebrafish coronary vessels with nerve fibers in the heart chamber, but coronary sprouts are not tightly associated with nerves.

We then examined the association of coronary sprouts with compact muscle, a tissue that is gradually vascularized during

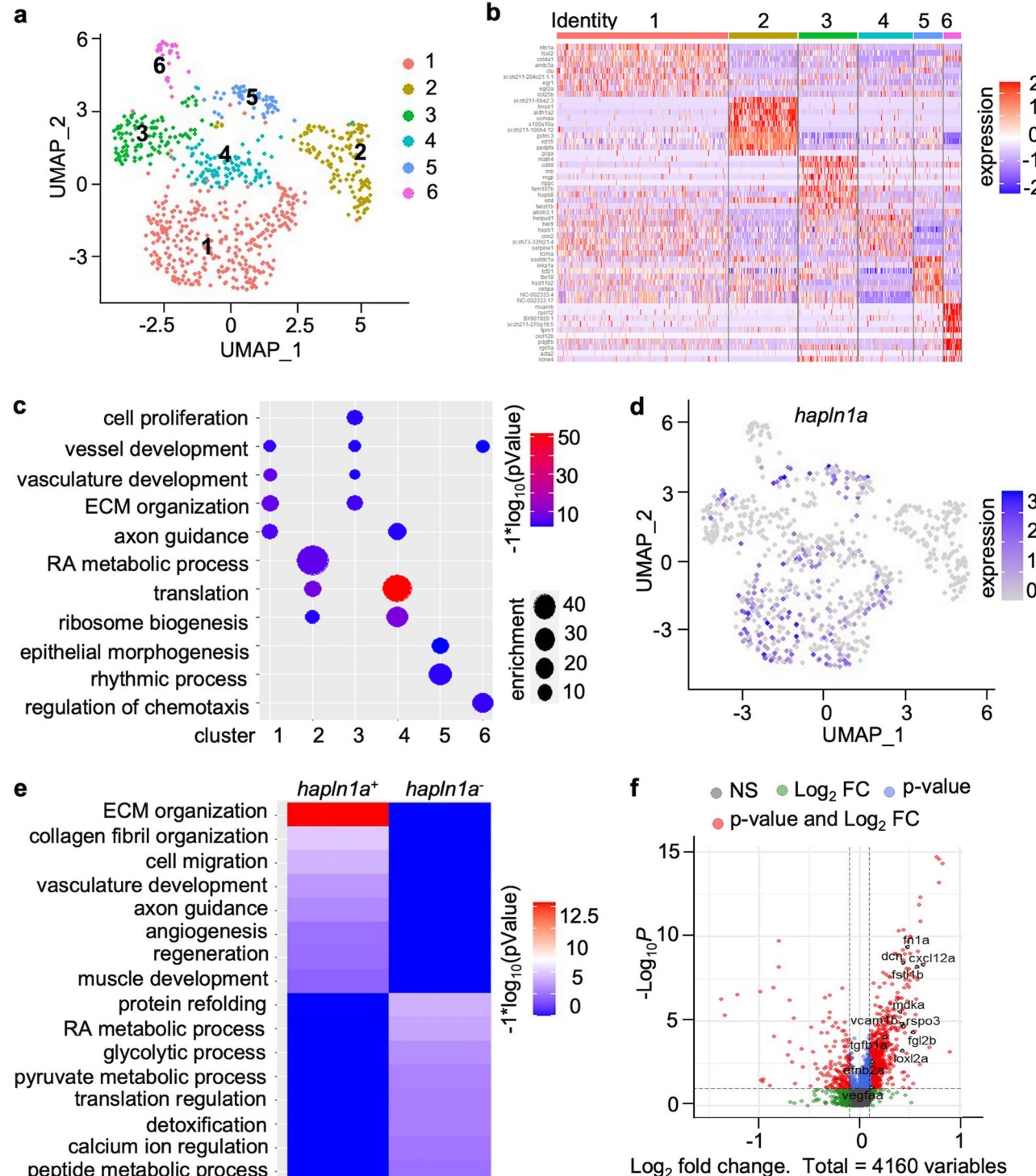

**Fig. 1 | Single-cell RNA-sequencing reveals hapln1a+ epicardial clusters are enriched with angiogenesis factors during coronary vascularization. a** Uniform manifold approximation and projection (UMAP) clustering of *tcf21*+ single-cells from juvenile hearts. **b** Heatmap of the top 10 markers for epicardial cells from juvenile *tcf21:nucEGFP* hearts. **c** Identification of epicardial cell clusters based on gene ontology analysis. **d** Feature plot of *hapln1a* expression in epicardial clusters of juvenile hearts. **e** Heatmap of gene ontology enrichment for *tcf21*+/*hapln1a*+ epicardial cells vs. *tcf21*+/*hapln1a*− cells. **f** Log2-fold-change vs. abundance of normalized gene expression of *tcf21*+/*hapln1a*+ epicardial cells vs. *tcf21*+/*hapln1a*− cells. Each point designates a unique gene; differentially expressed unique genes are in red. Unique genes with higher expression in *tcf21*+/*hapln1a*+ epicardial cells have positive fold-change values, whereas unique genes with higher expression in *tcf21*+/*hapln1a*− cells have negative fold-change values.

myocardial compaction. Zebrafish heart development starts within 24 h post-fertilization and continues throughout both juvenile and early adult life stages. In juvenile zebrafish, some *gata4*:EGFP+ CMs initially appear on the ventricular surface at ~5 weeks of age and

gradually expand and eventually encapsulate the whole chamber to create a contiguous wall of compact muscle[62,63]. Recent reports suggest coronary vessels can form a scaffold for cardiomyocyte production[12], but whether expanded cardiomyocytes affect coronary

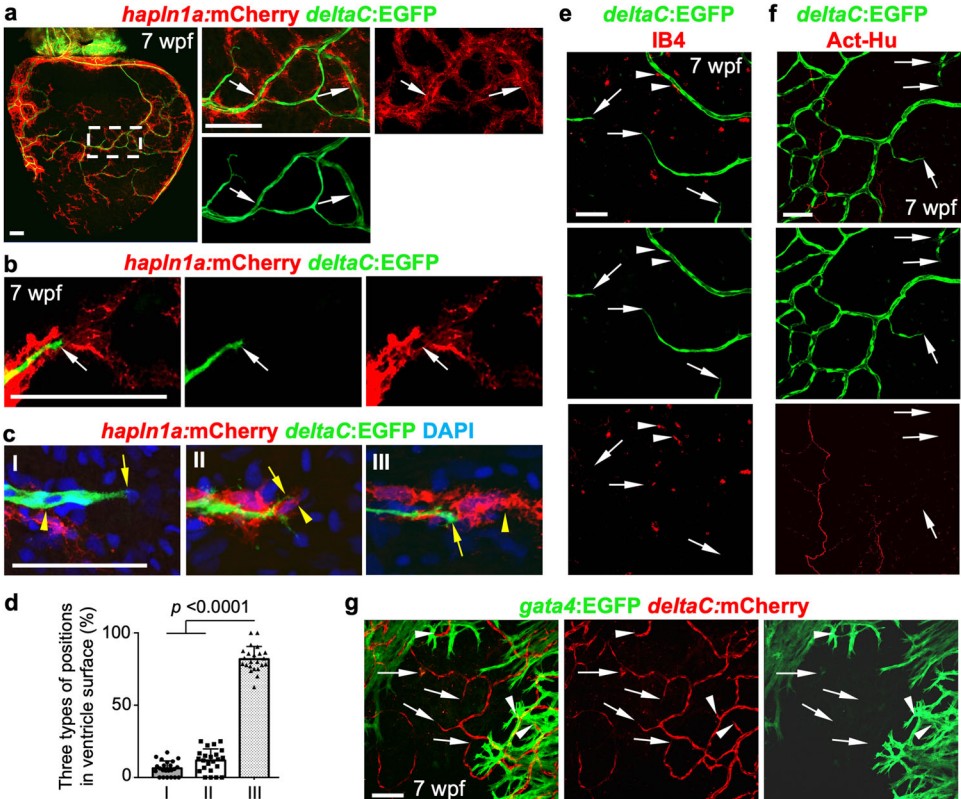

**Fig. 2 | hapln1a+ cells form cellular structures in advance of coronary sprouts during vascularization. a** Whole-mount view of *hapln1a*+ cells and *deltaC*:EGFP+ coronary endothelial cells on *hapln1a:mCherry-NTR;deltaC:EGFP* ventricular surface (opposite the atrioventricular junction) at 7 weeks post-fertilization (wpf). The boxed area is enlarged. Arrows indicate *deltaC*:EGFP+ cells lining with *hapln1a*+ cells. *n* = 50 animals. Scale bar, 100 μm. **b** Representative images of *hapln1a*+ cells located ahead of *deltaC*:EGFP+ cells in the juvenile ventricular surface. Arrows indicate coronary tips. Total 50 animals have been examined from 3 independent experiments with similar results. Scale bar, 50 μm. **c** The three position types of *hapln1a*+ cells according to their localization with coronary sprouts in the juvenile heart surface: *hapln1a*+ cells behind (I), parallel (II), and leading (III) *deltaC*+ cells. Arrows, front of *deltaC*+ cells. Arrowheads, front of

*hapln1a*+ cells. Scale bar, 50 μm. **d** Quantification of the three types of positions from experiments in **c**. I, 6%; II, 12%; III, 82%. *n* = 23 animals. Mann–Whitney rank-sum test (Two-sided). Data are presented as mean values ± SD. Source data are provided as a Source Data file. **e** Growing *deltaC*+ cell extensions and macrophages (IB4+) in juvenile hearts. Arrowheads represent IB4+ macrophages flanking coronary vessels. Arrows represent *deltaC*+ vessel extensions. *n* = 8 animals. Scale bar, 50 μm. **f** Growing *deltaC*+ cell extensions and AcT-Hu+ nerves in juvenile hearts. Arrows represent EGFP+ vessel extensions. *n* = 5 animals. Scale bar, 50 μm. **g** Growing *deltaC*+ cell extensions and *gata4*+ CMs in juvenile hearts. Arrows represent *deltaC*+ vessel extensions outside *gata4*+ CM area. Arrowheads represent *deltaC*+ vessel extensions embedded in *gata4*+ CM area. *n* = 8 animals. Scale bar, 50 μm.

growth has not been examined. We assessed juvenile *gata4:EGFP;deltaC:mCherry* hearts[47] and observed mCherry+ signals embedded in the EGFP+ area on the ventricular surface. However, most mCherry+ cell extensions localized outside of the EGFP+ area, indicating coronary sprouts are not tightly associated with expanding cardiomyocytes (Fig. 2g).

Overall, our results indicate that coronary growth has no tight correlation with macrophages, nerves, and cardiac muscles during vascularization.

## Coronary vessels grow along pre-formed *hapln1a*+ cellular structures

Given our observations on the interaction of coronary sprouts with *hapln1a*+ epicardial cells, macrophages, nerves, and cardiomyocytes, we hypothesized that the pre-formed *hapln1a*+ cell structure is instrumental to promoting coronary growth. To test this possibility, we first utilized an in vitro system to examine the migration of *hapln1a*+ cells and coronary endothelial cells. In this system, extracted and cultured heart tissue adhere to the base of a dish after 2–3 days and then cardiac tissues migrate outside of the cultured heart tissue. We extracted and cultured juvenile *hapln1a:mCherry;deltaC:EGFP* heart tissue in a dish. Daily imaging showed that both mCherry+ cells and EGFP+ cells migrated away from the cultured hearts and spread to the

dish bottom. Interestingly, *hapln1a*+ cells were always located in front of *deltaC*+ cells (Fig. 3a), suggesting a leading role of *hapln1a*+ cells for coronary extension.

Next, we examined *hapln1a*+ cells and coronary growth in an ex vivo system, which recently reported visualizing robust coronary expansion on the surface of juvenile fish ventricles within several days[47]. When first culturing juvenile *hapln1a:mCherry;deltaC:EGFP* hearts, mCherry+ linear structures are already established on the ventricular surface with no EGFP green signals; 6 h later, green signals protruded from existing EGFP tubules and extended along the established red linear structure; 12 h later, green signals continued to extend and bifurcate following the mCherry linear structure (Fig. 3b). Interestingly, we also observed coronary extension in an area without *hapln1a*+ cells as it had no detectable mCherry+ signals, but this growth was transient and quickly withdrawn (Fig. 3b). These results indicate coronary growth is guided by pre-formed *hapln1a*+ cellular structures and coronary extension may be transient without *hapln1a*+ cells during vascularization.

## *hapln1a*+ cells are required for coronary vascularization

To further identify the requirement of *hapln1a*+ cells for coronary vascularization, we examined coronary growth under the condition of *hapln1a*+ cell loss with *hapln1a:mCherry-NTR* fish[23]. First, we examined

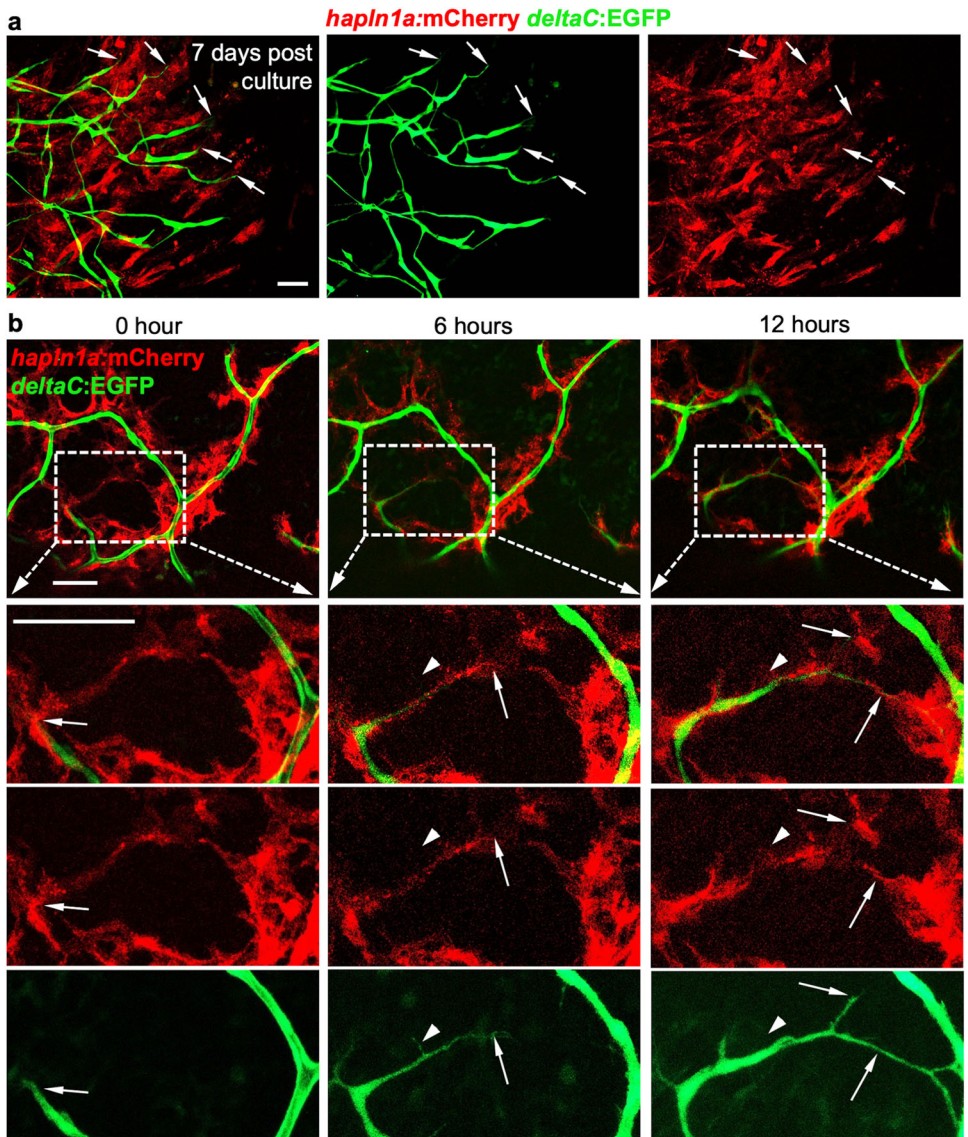

**Fig. 3 | Coronary vessels grow along pre-formed hapln1a+ cellular structures.**
**a** Visualization of the growth of *deltaC*:EGFP⁺ and *hapln1a*:mCherry⁺ tissues from *deltaC:EGFP;hapln1a:mCherry* hearts over 7 days after in vitro culture. *deltaC*:EGFP⁺ tissue growth is clearly lagging behind *hapln1a:mCherry*⁺ cells. Arrows represent growing coronary extensions. Total 5 tissue aggregates were examined. The experiments were repeated two times independently with similar results. Scale bar, 50 μm. **b** Live imaging of *hapln1a*⁺ and *deltaC*⁺ cells in the ventricular surface of ex vivo juvenile hearts by 0, 6, and 12 h, respectively. Total 12 hearts were observed from three independently experiments with similar results. Visualization of *deltaC*:EGFP⁺ coronary growth following existing *hapln1a*⁺ cell shears is clear (32 out of 39 observed extension growth). Arrows represent growing coronary tips. Arrowheads represent growing *deltaC*⁺ tips lacking *hapln1a*⁺ linear cell structures. Scale bar, 50 μm.

ex vivo coronary growth in juvenile ventricles after *hapln1a*⁺ cell depletion[47]. The 7 wpf juvenile *hapln1a:NTR;deltaC:EGFP* fish and *deltaC:EGFP* siblings were treated with 10 mM Metronidazole (Mtz) for 12 h each day for 2 continuous days. Then, the hearts were extracted and cultured ex vivo. Daily imaging revealed that EGFP⁺ cells did not extend in *hapln1a:NTR;deltaC:EGFP* hearts with *hapln1a*⁺ cell depletion, compared with robust coronary growth in *deltaC:EGFP* hearts (Fig. 4a, b). Next, we evaluated in vivo coronary vascularization in juvenile fish after *hapln1a*⁺ cell ablation. The 7 wpf *hapln1a:NTR;deltaC:EGFP* fish and *deltaC:EGFP* siblings were first treated with 10 mM Mtz for 12 h each day for 2 continuous days and returned to the fish aquarium. Then, we performed the third treatment at 5 days after the first treatment. At 5 days after the third Mtz treatment, we collected hearts and assessed coronary vessels in the ventricles. The coronary vessel density and junctions in *hapln1a:NTR;deltaC:EGFP* animals were significantly lower than in *deltaC:EGFP* sibling controls

(Fig. 4c–e), while coronary vessel numbers in *hapln1a:NTR;deltaC:EGFP* animals were more than in *deltaC:EGFP* sibling controls (Fig. 4f). As *hapln1a*⁺ cells also envelop existing coronary vessels and behave as perivascular cells, these results indicate that coronary growth is not only blocked but coronary vessels also become unstable without *hapln1a*⁺ cells. This instability caused the fragmentation of existing vessels and resulted in the increased vessel numbers in the condition of *hapln1a*⁺ cell loss.

### *hapln1a*⁺ cells are required for coronary revascularization during regeneration

Next, we asked if *hapln1a*⁺ cells play a role in coronary regrowth during heart regeneration. We examined *hapln1a*⁺ cells and coronary vessels in the regenerating injury area and observed coronary sprouts closely associated with pre-formed *hapln1a*⁺ cell structures (Fig. 5a, b). To test the requirement of *hapln1a*⁺ cells for coronary revascularization, we

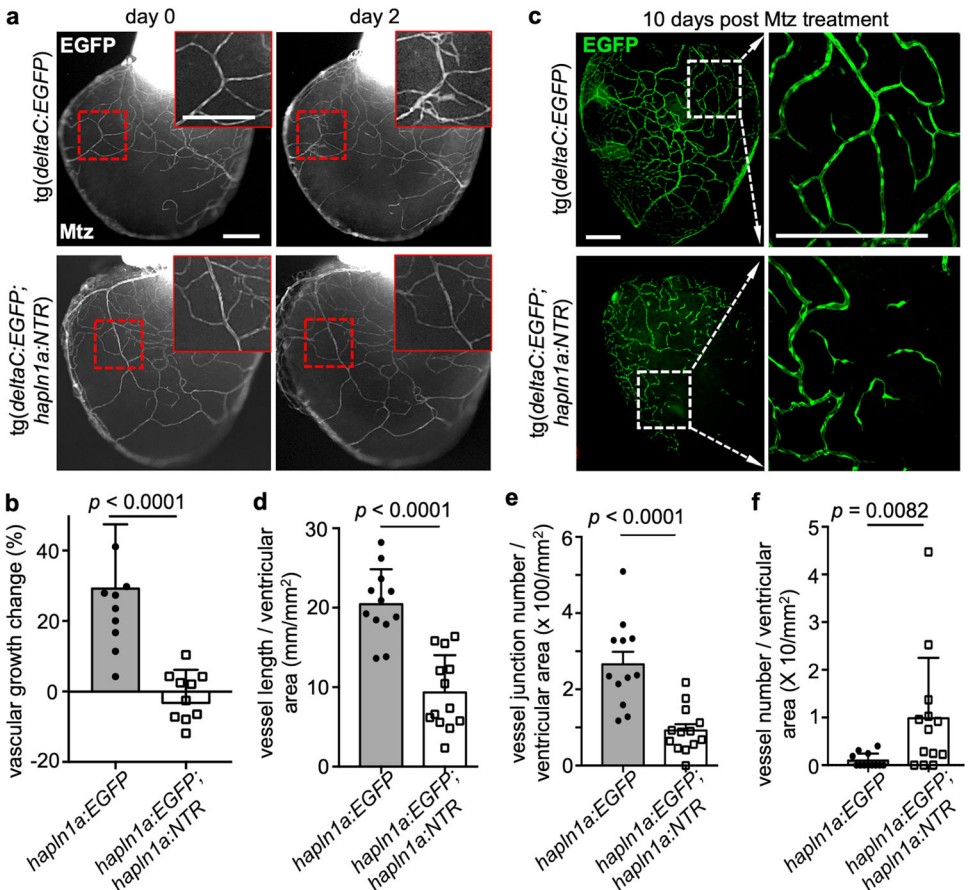

**Fig. 4 | hapln1a+ cell depletion resulted in defective coronary vascularization.**
**a** Obvious ex vivo *deltaC*⁺ cell growth (red boxes included) within 2 days in control *deltaC:EGFP* hearts and no growth in *deltaC:EGFP;hapln1a:NTR* hearts, after Mtz treatment. The experiment was repeated once. Scale bar, 200 μm. **b** Quantification of pixels of EGFP⁺ signals at day 2 versus that at day 0 in experiment (**a**). *n* = 11 in each group. Mann–Whitney rank-sum test (two-sided). Data are presented as mean values ± SD. Source data are provided as a Source Data file. **c** Whole-mount view of the ventricular surface from in vivo juvenile *deltaC:EGFP;hapln1a:NTR* (*n* = 13) and *deltaC:EGFP* (*n* = 12) clutchmates at the age of 8 wpf that were treated with 10 mM

Mtz for 12 h at day 0, day 1, and day 5, and then the hearts were extracted at day 10 and analyzed. The experiment was repeated once. Scale bars, 200 μm.
**d**–**f** Quantification of pixels of EGFP⁺ signals (**d**), vessel junctions (**e**), and isolated vessels (**f**) versus whole ventricular area in experiment **c**. In the quantifications in **d**, **e**, **f**, *n* = 13 juvenile *deltaC:EGFP;hapln1a:NTR* fish and *n* = 12 juvenile *deltaC:EGFP* fish. The experiment was repeated once with similar results. d, e, f, Mann-Whitney Rank Sum test (Two-sided). Data are presented as mean values ± SD. Source data are provided as a Source Data file.

---

utilized the *hapln1a:NTR* fish strain to deplete *hapln1a*⁺ cells during heart regeneration. Adult *hapln1a:NTR;deltaC:EGFP* animals and *deltaC:EGFP* siblings were treated with 10 mM Mtz from 2 days post-amputation (dpa) for 12 h per day for 3 continuous days, and hearts were collected at 7 dpa. We determined that coronary vessel density in the regenerating area is significantly lower in *hapln1a:NTR;deltaC:EGFP* fish hearts, compared with *deltaC:EGFP* controls (Fig. 5c, d). These results implicate the requirement of *hapln1a*⁺ cells for coronary revascularization during heart regeneration. As *hapln1a*⁺ cells form around 30% of *tcf21*⁺ cells during morphogenesis (Fig. 1d) and around 50% during regeneration[23], we also examined the effect of randomly ablating around 50% of *tcf21*⁺ cells (Supplementary Fig. 5a, b). With this method of *tcf21*⁺ cell ablation, there was no significant difference in coronary vascularization and revascularization when compared with control animals (Supplementary Fig. 5c–f), indicating that ablating *hapln1a*⁺ cells has a more severe effect on coronary growth.

**Locally expressed *serpine1* in *hapln1a*⁺ cells regulates coronary growth**
To understand the molecular nature of *hapln1a*⁺ cells on guided coronary growth, we performed scRNA-seq analyses with *hapln1a*⁺ cells from juvenile and regenerating hearts. The *hapln1a*⁺ cells were isolated from the ventricles of juvenile *hapln1a:EGFP* animals at 7 wpf and the

regenerating area of the ventricular apex of adult animals at 7 dpa. A total of 4419 *hapln1a*⁺ cells from juvenile hearts and 2637 *hapln1a*⁺ cells from adult hearts were sequenced and analyzed. After examining gene expression in these cells, we found that *hapln1a*⁺ cell clusters not only highly express genes with known roles in vascular growth, such as *vegfaa*, *cxcl12a*, *ccbe1*, *tgfb1a*, and *cpn1.1* (Supplementary Fig. 6)[16,24,64–68], but they also express the *serpine1* gene (also known as plasminogen activator inhibitor-1), which was recently implicated in cardiomyocyte proliferation during zebrafish heart regeneration[69]. Previous studies demonstrated that *serpine1* regulates neuron migration during development in mice and humans and mediates cancer invasion and metastasis, through modulating interactions between cells and the extracellular matrix (ECM) to control the balance between cell adhesion and migration[70,71]. Therefore, we speculated that *hapln1a*⁺ cells may regulate coronary growth through *serpine1*.

To test this possibility, we first performed in situ hybridization analyses and confirmed *serpine1* expression in *hapln1a*⁺ cells during coronary vascularization and revascularization (Fig. 6a–c). We found that *serpine1* signals closely localized to coronary sprouts (Fig. 7a–c), while other genes like *vegfaa*, *cxcl12a*, *ccbe1*, *tgfb1a*, and *cpn1.1* did not show a close correlation with coronary sprouts (Supplementary Fig. 7), suggesting that *hapln1a*⁺ cell-derived *serpine1* may regulate coronary growth. Therefore, we next examined coronary growth

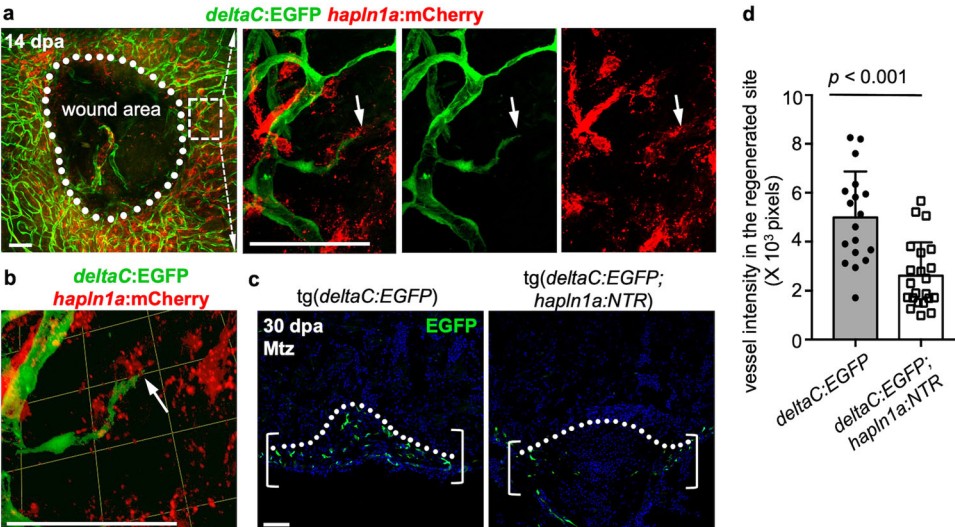

**Fig. 5 | Coronary re-growth follows and requires hapln1a+ cell structures.**
**a** Whole-mount view of coronary vessels and *hapln1a*⁺ cells in the wound area of adult injured *deltaC:EGFP;hapln1a:mCherry* animals. White dashed box is enlarged in single confocal slices. Arrows label the coronary tips. White dashed line circles the closing wound. *n* = 10 animals. Scale bars, 50 μm. **b** 3-D visualization of coronary sprouts and *hapln1a*⁺ cells in the regenerating area in experiment (**a**). Arrow represents the coronary tip lagging behind *hapln1a*⁺ cells. Scale bars, 50 μm.
**c** Section views of coronary vessels in the wounded area in adult injured *deltaC:EGFP;hapln1a:mCherry* animals and *deltaC:EGFP* siblings, with Mtz treatment by 30 dpa. Brackets represent the regenerated area. Scale bars, 50 μm.
**d** Quantification of EGFP⁺ pixels in the regenerating site from (**c**). *n* = 18 *deltaC:EGFP* and *n* = 22 *deltaC:EGFP;hapln1a:mCherry* animals were used. The experiments were repeated at least once. Mann–Whitney rank-sum test (Two-sided). Data are presented as mean values ± SD. Source data are provided as a Source Data file.

after blocking *serpine1* function under different conditions by treatment with its antagonist Tiplaxtinin, which blocks the Serpine1 protease and has been used to inhibit *serpine1* activity in different systems, including during zebrafish heart regeneration[69,72–76]. Previous studies measured and determined that the inhibitory effect of 20 μM of Tiplaxtinin on *serpine1* is ~95%[72,77]. We first examined ex vivo juvenile ventricular coronary growth after blocking *serpine1* function with 20 μM Tiplaxtinin and observed blocked coronary growth in the dish (Fig. 8a, b). Moreover, these blocked coronary extensions can re-start after drug removal, indicating that the coronary growth restriction by *serpine1* inhibition does not occur by coronary cell death (Supplementary Fig. 8). Next, we treated 6 wpf juvenile fish with Tiplaxtinin for 12 h and continuously for 7 days. We found fewer coronary vessels in hearts treated with Tiplaxtinin when compared with vehicle controls (Fig. 8c, d). Lastly, we treated adult fish with Tiplaxtinin from 3 dpa for 12 h and continuously for 4 days and observed fewer coronary vessels in the regenerating area at 7 dpa, compared with vehicle controls (Fig. 8e, f). Similar to a previous report that the endocardium expresses *serpine1* after heart injury[69], we also detected *serpine1* signals in the inner injury area. However, we speculate that these *serpine1*-expressing cells are not directly correlated with coronary extension as coronary vessels mainly extended from the lateral area of the ventricular wall to the middle of the injury site. Together, our results indicate that *hapln1a*⁺ cell-derived *serpine1* controls coronary growth during heart morphogenesis and regeneration.

## Depletion of *hapln1a*-expressing cells or inhibition of *serpine1* activity results in defective hyaluronic acid deposition near coronary sprouts

To elucidate the mechanism by which *hapln1a*⁺ cells and *serpine1* activity impact coronary growth, we assessed the distribution of the *hapln1a*⁺ cell substrate, hyaluronic acid (HA). We recently reported that *hapln1a*⁺ epicardial cells regulate HA deposition to facilitate cardiomyocyte proliferation during heart morphogenesis and regeneration[23]. HA is required for coronary revascularization during zebrafish heart regeneration[78–80]. Previous studies also demonstrated

that endothelial cells can attach and grow along HA fibers in vitro, and a scaffold in tissue-engineered vascular grafts formed from HA-based biomaterials can completely generate a new vascular tube. These studies indicate that the administration of HA can enhance cell proliferation, adhesion, tubular sprout formation, and the migration of endothelial cells[78–84]. We speculate that HA is involved in guided coronary growth. To examine this possibility, we first assessed HA localization with coronary vessels on the ventricular surface in juvenile *deltaC:EGFP* animals. We detected strong association of linear HA signals with EGFP⁺ signals (Fig. 9a). Next, we examined the HA deposition after ablating *hapln1a*⁺ cells at 7 wpf. Juvenile *hapln1a:mCherry-NTR;deltaC:EGFP* animals and *deltaC:EGFP* siblings were treated with 10 mM Mtz and their hearts were collected for histological analysis. We assessed the size of HA aggregates by quantifying the area of each focus of HA and found that *hapln1a*⁺ cell-depleted hearts displayed 35% lower HA intensity per HA aggregate, when compared with control siblings (Fig. 9b), indicating that the ablation of *hapln1a*⁺ cells disrupted the linear structure of HA in the coronary growth area. Next, we examined HA signals in the regeneration area and found HA signals were closely associated with *deltaC:EGFP*⁺ cells (Fig. 9c). We then assessed the HA deposition within the regenerating area after depleting *hapln1a*⁺ cells and observed that 7 dpa injury sites of *hapln1a*⁺ cell-depleted animals displayed ~47% smaller puncta of HA, compared with wild-type siblings (Fig. 9d). Next, we assessed whether *serpine1* functions through regulating the HA structure during morphogenesis and regeneration, as previous reports indicated *serpine1* protects the ECM to maintain a matrix necessary for endothelial cells to migrate and form capillaries[85]. We examined HA deposition in the ventricle surface after treating juvenile *deltaC:EGFP* fish with Tiplaxtinin for 12 h and continuously for 7 days and found that inhibitor treated hearts displayed a 64% lower HA intensity per HA aggregate, when compared with control siblings (Fig. 9e, f), revealing a disorganized HA linear structure within the coronary growth area. Lastly, we treated adult fish with Tiplaxtinin from 3 dpa for 12 h and continuously for 4 days and observed 53% lower HA intensity per HA aggregate at 7 dpa, compared with vehicle controls (Fig. 9g, h). Our results indicate a mechanism for *hapln1a*⁺ cell function during morphogenesis and regeneration, in

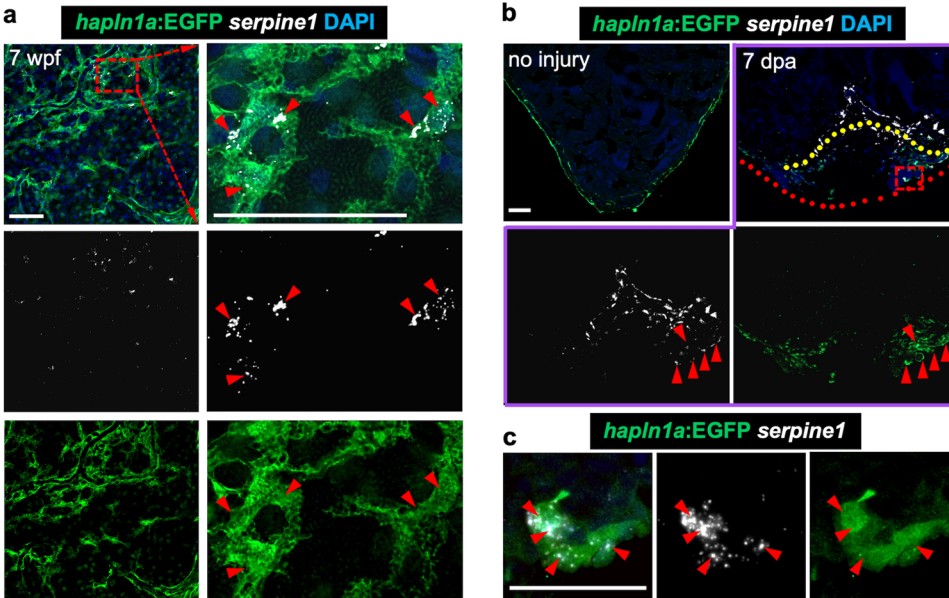

**Fig. 6 | hapln1a+ cells adjacent to coronary sprouts express the serpine1 gene. a** Visualization of *hapln1a:EGFP+* cells and in situ signals of *serpine1* mRNA in whole-mount view of the ventricular surface of juvenile *hapln1a:EGFP* animals. Red dashed rectangle is enlarged. Red arrowheads indicate in situ signals, located in *hapln1a+* cells. *n* = 6. Scale bar, 50 μm. **b** In situ of *serpine1* mRNA in section views of adult injured *hapln1a:EGFP* hearts. Red arrowheads indicate in situ signals distributed in EGFP+ cells, which are near the area of coronary vessel regrowth. Yellow dashed lines represent surgery plane. Red dashed lines outline ventricular apex. Red rectangle area is enlarged in **c**. *n* = 5. Scale bar, 50 μm. **c** Representative image of *serpine1* in situ signals located in EGFP+ cells in the wound area. Red arrowheads indicate in situ signals. Scale bar, 25 μm.

which HA is deposited by *hapln1a+* cells and then organized by *serpine1*-expressing cells for coronary growth during vascularization and revascularization.

## Discussion

In this study, we profiled *tcf21+* epicardial cells from juvenile zebrafish hearts and characterized the role of *hapln1a+* epicardial cells in coronary growth during heart morphogenesis and regeneration. Our work demonstrates that coronary growth is a *hapln1a+* cell-guided process and identifies *serpine1* as a critical regulator for coronary growth.

The epicardium is essential for heart development and regeneration. Transcriptional profiles of the epicardium and epicardium-derived cells were reported in embryonic, adult, and regenerating hearts[86–89]. Recently, we demonstrated that *hapln1a+* epicardial cells define epicardial subpopulations for cardiomyocyte proliferation during heart morphogenesis and regeneration[23]. Here, our results indicate coronary formation is a key function of *hapln1a+* epicardial cells. In addition to its role in wrapping and stabilizing newly formed coronary vessels, *hapln1a+* cells precede vascular sprouts to guide coronary growth. This process may be analogous to guidepost cell populations that can be transient and act to assist biological events. For example, guidepost cell populations provide a scaffold or release paracrine factors during axon targeting[90,91]. The observation of coronary growth following and relying on *hapln1a+* cells suggests that *hapln1a+* cells ostensibly behave as a cellular cue of a guiding scaffold. Epicardial cells are also known to secrete developmental factors, including many ECM components, during heart regeneration[22,23,92]. Our work indicates that *hapln1a+* cells regulate HA organization during heart morphogenesis and regeneration. Although HA has previously been implicated in vessel growth stimulation and been applied during bioengineering for vessel tube formation[69,78–84], no studies have reported that HA lines coronary vessels and paves the road for coronary growth during morphogenesis and natural regeneration. These *hapln1a+* cell-derived HA cables may function as scaffolds, which could prevent loss of ECM components during tissue remodeling, act as a template for matrix regeneration, and support interactions with other cell types[93–97]. Our recent and current studies raise an intriguing possibility that *hapln1a+* epicardial cells oversee principal cardiogenic activities including myocardial expansion and coronary growth. In future, understanding the molecular nature of *hapln1a+* subpopulations on cardiomyocyte proliferation and coronary growth, and how they control local cardiogenesis can inspire new ideas for targeted tissue regeneration.

Our work on the function of *serpine1* on coronary growth also provides new evidence of epicardial cells regulating the regeneration environment and further demonstrates that nerves and vessels share similar mechanisms to regulate their migration[98]. Our previous report indicated that the epicardial cluster 1 not only deposits HA through *has1* expression but also requires *hapln1* to stabilize HA structures[23]. Our current results indicated that *serpine1* is involved in this process by locally organizing linear HA structures to guide coronary extension. In this study, we discovered that *hapln1a+* cells and *serpine1* activity are responsible for major coronary growth in zebrafish: *hapln1a+* cells deposit HA and *serpine1* regulates the HA organization to form a linear structure, which is required for coronary extension and continuous growth. As *hapln1* expression is distributed in the cells wrapping and in advance of coronary vessels, we speculate that the hapln1 protein plays a general role in organizing HA structure by stabilizing the binding of HA with other ECM proteins like proteoglycan, but it likely does not play a major role in the guidance of coronary growth as it is not expressed locally in the area of coronary extension. Further, our analyses also revealed that there is a small percentage (around 18%) of coronary growth extensions that occur without any *hapln1a+* cells during heart morphogenesis. Although we also observed the withdrawal of the extended coronary vessels without *hapln1a+* cells, we cannot exclude other mechanisms that provide guidance cues that may affect coronary growth during vascularization and revascularization. Further, we noticed that the percentage of *serpine1* expression in *hapln1a+* cells is different during heart morphogenesis and regeneration. We speculate that this difference arose because coronary vascularization occurs for around two months (from 5 wpf −12 wpf) and builds the coronary vasculature from very few coronary cells, while

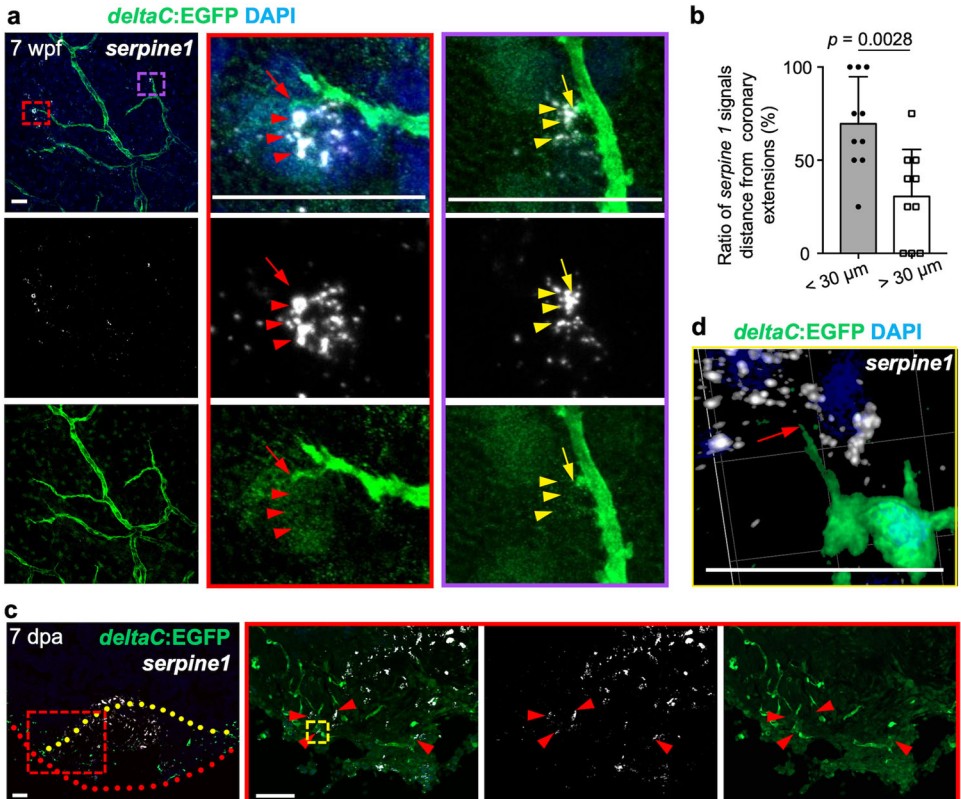

**Fig. 7 | hapln1a+ cells adjacent to coronary sprouts express the serpine1 gene.**
**a** Visualization of EGFP+ cells and in situ signals of *serpine1* mRNA in whole-mount view of the ventricular surface of juvenile *deltaC:EGFP* animals. Red dashed rectangle is enlarged. Red arrow indicates coronary sprouts and arrowheads indicate in situ signals. The *serpine1* in situ signals are located laterally and ahead of EGFP+ signals. *n* = 7. Scale bar, 50 μm. **b** Quantification of the two types of positions: *serpine1* signals within 30 μm. of coronary extensions and more than 30 μm, from experiments in **a**. *n* = 10 animals. The 69% quantified coronary extensions have *serpine1* signals within 30 μm in radius, and 31% of counted coronary extensions

have no *serpine1* signals within 30 μm. Mann–Whitney rank-sum test (two-sided); made for two comparisons. Data are presented as mean values ± SD. Source data are provided as a Source Data file. **c** In situ of *serpine1* mRNA in section views of adult injured *deltaC:EGFP* hearts. Red arrowheads indicate in situ signals. Yellow dashed lines represent surgery plane. Red dashed lines outline ventricular apex. Yellow rectangle area is enlarged in **d**. Red arrowheads label *serpine1* in situ signals in the wound edge. *n* = 5. Scale bar, 50 μm. **d** 3-D view of *serpine1* in situ signals and *deltaC*:EGFP+ cells in the wound area. Red arrow indicates the coronary tip located in the area distributed with in situ signals. Scale bar, 200 μm.

coronary revascularization occurs within 2–3 weeks and generates new vessels from existing vessels. Another possibility is that the endocardium functions through *serpine1* to regulate myocardial regeneration while such endocardium-derived *serpine1* doesn't exist during heart morphogenesis. We speculate that these differences led us to detect low expression levels of *serpine1* in morphogenesis hearts in comparison with *serpine1* expression in the injury site. Further, a lower disruption effect on the HA structure was observed after blocking *serpine1* in juvenile hearts when compared with other conditions such as depleting *hapln1a*+ cells in juvenile and regenerating hearts and blocking *serpine1* in regenerating hearts. One reason for these variations is that *hapln1a*+ cell loss not only leads to small aggregates of HA but also less HA production[23], while *serpine1* plays an organizational role in the local coronary extension area. Further, although the heart regeneration process always recapitulates the developmental process, coronary vascularization in juvenile fish takes around 2 months (from 5 wpf to 12 wpf), while revascularization occurs within 2–3 weeks. This temporal difference may cause different speeds of ECM deposition and coronary growth during development and regeneration. Accordingly, blocking *serpine1* during development may cause a lower disruption effect on HA structure in juvenile hearts when compared with regenerating hearts.

Previous studies implicated other cell types in the process of guiding coronary growth. For example, macrophages localize near coronary tips during embryonic mice heart development[14]. In our

study, we observed macrophages attached to coronary vessels, but the phenomenon of macrophages associating with coronary growth tips was seldom observed. We speculate that macrophages may regulate zebrafish coronary vessel sprouting, but they most likely do not play a major role in guiding coronary growth as they are not tightly associated with one another. Cardiomyocytes have also been reported to secrete mitogens like *cxcl12* to regulate the vascularization of zebrafish compact muscle[16]. We examined proliferating cardiomyocytes and coronary vessels during myocardial compaction and observed coronary growth sprouts outside of the area of expanding compact myocardium, indicating that coronary growth is mainly not regulated by newly formed myocardium. Further, we examined nerves and coronary sprouts during coronary vascularization and didn't observe a tight correlation between coronary sprouts and nerves. Compared with these current proposed guiding cells, our study reveals that *hapln1a*+ cells may play a major role in guiding coronary growth.

## Methods
### Zebrafish and heart injuries
All animal procedures were approved by the Institutional Animal Care and Use Committee of Emory University and performed in accordance with Emory University guidelines. 4-10-month-old outbred EK or EK/AB zebrafish were used for ventricular resection surgeries[7]. All animals were used with male to female at 1:1 when sex determination has occurred. To deplete *hapln1a*-expressing cells,

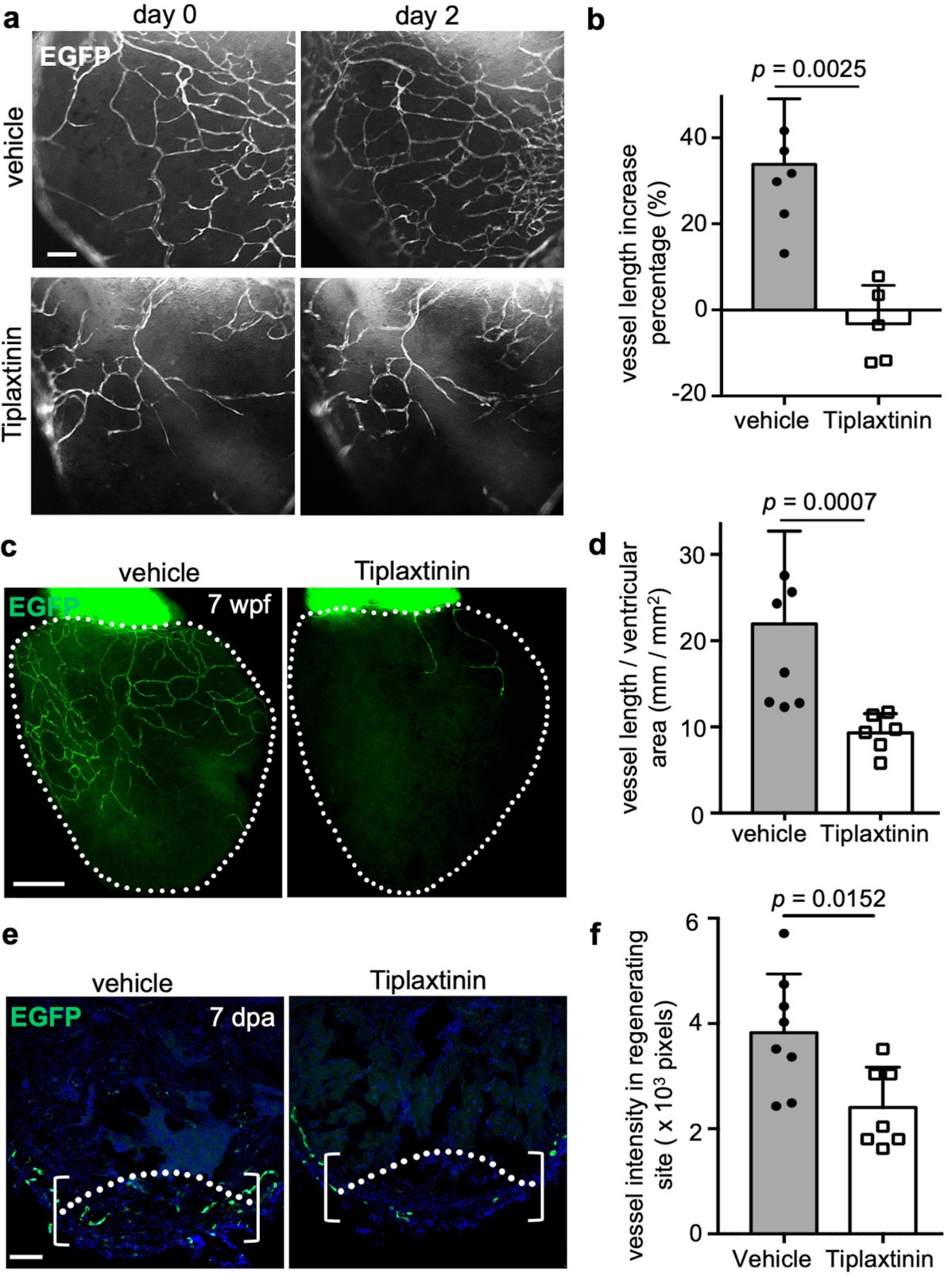

Tg(*hapln1a:mCherry-NTR*)[em14Tg 23] transgenic zebrafish were used at the age of 6-7 weeks with a length between 1.5 and 1.8 cm (juvenile), or 4–10 months with a length of at least 3 cm (adult). Animal density was maintained at ~4 fish/L in all experiments. Animals were randomly assigned to the test groups in all experiments. To deplete as many *hapln1a*[+] cells as possible and avoid non-specific toxic effects, juvenile and adult *hapln1a:mCherry-NTR* animals and their control siblings were treated with vehicle or 10 mM Mtz in different conditions: fish were incubated with vehicle/Mtz in 1.5 L mating tanks for 12 h/day, and the treatment continued for 2 days in juvenile fish and 3 days in adult fish. Transgenic strains described elsewhere include: Tg(*gata4:EGFP*)[ae1Tg99], Tg(*tcf21:nucEGFP*)[pd41Tg99], Tg(*hapln1a:EGFP*)[pd338Tg23], Tg(*hapln1a:mCherry-NTR*)[em14Tg23], Tg(*deltaC:EGFP*)[em11Tg47], Tg(*deltaC:mCherry*)[em11Tg47]. All transgenic strains were analyzed as hemizygotes.

**Fig. 8 | Inhibiting serpine1 function blocks coronary growth during vascularization and revascularization. a** Daily imaging of EGFP⁺ coronary cells in juvenile *deltaC:EGFP* ventricular surface treated with *serpine1* inhibitor ($n = 7$) or vehicle treatment ($n = 5$) ex vivo. The experiment was repeated once. Scale bars, 50 μm. **b** Quantification of the percentage of EGFP⁺ pixels on the ventricular surface from experiments in **a**. The experiment was repeated once with similar results. Mann−Whitney rank-sum test (two-sided). Data are presented as mean values ± SD. Source data are provided as a Source Data file. **c** Visualization of EGFP⁺ coronary cells in whole-mounted juvenile *deltaC:EGFP* hearts at the age of 7 wpf after *serpine1* inhibitor ($n = 6$) and vehicle treatment ($n = 8$). The experiment was repeated once. White dashed lines, ventricle. Scale bars, 200 μm. **d** Quantification of the percentage of EGFP⁺ pixels on the ventricular surface from experiments in **c**. The

experiment was repeated once with similar results. Mann−Whitney rank-sum test (two-sided). Data are presented as mean values ± SD. Source data are provided as a Source Data file. **e** Section images of ventricles of vehicle- or *serpine1* inhibitor-treated *deltaC:EGFP* animals at 30 dpa, assessed for EGFP⁺ coronary cells in the injury site. $n = 8$ in vehicle control and $n = 7$ in serpine1 inhibitor-treated group. The experiment was repeated once. Brackets, injury site used for quantification. Dashed line indicates amputation plane. Scale bars, 50 μm. **f** Quantification of the percentage of EGFP⁺ pixels in the regenerating area from experiments in **e**. The experiment was repeated once with similar results. Mann−Whitney rank-sum test (two-sided). Data are presented as mean values ± SD. Source data are provided as a Source Data file.

## Histology

Histological analyses were performed on 10 μm cryosections or whole-mount paraformaldehyde-fixed hearts. For immunostaining of whole-mounted hearts or heart sections: samples were blocked with 2% horse serum, 1% DMSO, 10% heat inactivated new calf serum and 0.1% Tween-20 in PBS for 1 h at room temperature. Primary antibodies were diluted in PBS plus 1% DMSO, 10% heat inactivated new calf serum, and 0.1% Tween-20 and incubated with hearts overnight at 4 °C. Samples were then washed with PBS plus 0.1% Tween-20 and incubated with the secondary antibody diluted in the PBS plus 1% DMSO, 10% heat inactivated new calf serum, and 0.1% Tween-20 for 1 h at room temperature. Samples were mounted in Vectashield vibrance antifade mounting medium with DAPI (Vector Laboratories, H-1800-10). To determine the general innervation of the heart, we used antibodies against acetylated α-tubulin (AcT) combined with human neuronal protein C/D (HuC/HuD). Primary antibodies used in this study include IB4 (Vector labs, DL-1208-5, 1:100), HuC/HuD (Invitrogen, A21271, 1:100), AcT (Sigma-Aldrich, T6793-.2 ML, 1:100), and GFP (Aves Labs, GFP-1020, 1:200). Alexa Fluor secondary antibodies used include 488 (goat anti-Chicken, A-11039, 1:200) and 594 (goat anti-mouse, A-11005, 1:200). Whole-mounted and sectioned ventricular tissues were imaged using a Zeiss LSM800 confocal scanning microscope with ZEN V3.7.

The immuno-localization of HA was performed on paraformaldehyde-fixed hearts using a biotin-labeled HA-binding protein. Hyaluronic acid binding protein (HABP) recognizes HA saccharide sequences and is able to localize HA in tissues by streptavidin conjugation with an appropriate fluorophore[100]. To detect HA, samples were blocked with 2% BSA in PBS for 1 h at room temperature. HA was stained with HABP (Sigma, 385911, 2.5 μg/ml) in 3% BSA overnight at 4 °C. Samples were then washed with PBS and incubated with Alexa Fluor 633 streptavidin conjugate (Thermo Fisher, S21375, 1:500) in 3% BSA for 1 h. Samples were mounted in Vectashield vibrance antifade mounting medium with DAPI (Vector Laboratories, H-1800-10) and imaged using a Zeiss LSM800 confocal scanning microscope with ZEN V3.7.

### Fluorescence in situ hybridization

Probes were synthesized by T7 RNA Polymerase (Roche,10881767001) and SP6 RNA Polymerase (Roche, 0810274001) and labeled by Digoxigenin RNA labeling mix (Roche, 11277073910). Fluorescence in situ hybridization was performed as previously described[101]. In summary, hearts were removed and fixed overnight at 4 °C in 4% paraformaldehyde. After processing, these hearts were cryosectioned to obtain 10 μm sections and then hybridized with probes in hybridization buffer, followed by blocking and then incubation with anti-Digoxigenin-POD fab fragments (Roche, 11207733910) in blocking buffer. The mRNA was detected using a TSA Plus Cyanine 5 Kit (AKOYA BIOSCIENCES, NEL745001KT) according to the manufacturer's instructions. Next, the sections were processed with standard antibody staining protocols for immunofluorescence as described[7] and then

imaged. The primers for generating antisense mRNA are as follows: *serpine1* Forward = TTGGGCTACAGGTGTTTGCT and Reverse = GCC GGTCAAGTGTGATTTCC; *vegfaa* Forward = ATGAACTTGGTTGTTT ATTTGAT and Reverse = TCATCTTGGCTTTTCACATC; *cxcr12a* Forward = CGCCATTCATGCACCGATTT and Reverse = TGACCT-GATTCTGCTGAGCG; *ccbe1* Forward = TACCCGTGCGTAAAGTCCAC and Reverse = ACAGTCTCAAACCGGCCAAT; *tgfb1a* Forward = TGCTTGCTGGACAGTTTGGT and Reverse = GTGCCAACAGC TCGTCTCTT; and *cpn1.1* Forward = AAATGAGGTGCTCGGAAGGG and Reverse = TTTTAGCCGTCGATAGGCCC.

### Heart culture

The heart culture was performed as previously described[21,47]. In summary, hearts were rinsed several times in PBS after collection and embedded in 1% low melting agarose with DMEM medium plus 10% fetal bovine serum, 1% non-essential amino acids, 100 U/mL penicillin, 100 μg/mL streptomycin, 50 μg/mL Primocin™ (InvivoGen), and 50 μM 2-Mercaptoethanol. Hearts were placed on the gridded glass bottom of live imaging dishes, and cultured at 28 °C in an incubator with 5% $CO_2$. Tiplaxtinin (Selleckchem, S7922) was dissolved in ethanol to a stock concentration of 20 mM. Tiplaxtinin was used at a final concentration of 20 μM for in vivo treatment and ex vivo culture. Fluorescent transgenes in these hearts were monitored using a fluorescence stereomicroscope SteREO Discovery with ZEN V3.5 and Zeiss LSM800 confocal scanning microscope with ZEN V3.7.

### Coronary quantification in juvenile and regenerating hearts

To quantify coronary expansion in juvenile hearts from *deltaC:EGFP* fish, whole-mounted specimens were selected and ventricle images were captured using a 10x objective lens (1024 × 1024 pixels). The coronary vessel lengths were measured by ImageJ V1.53. To quantify the vessel sprouting activity, the total number of vessel junctions (branches ≥ 2) were counted. To quantify coronary vessel number in juvenile hearts from *deltaC:EGFP* fish, whole-mounted images of the ventricle were captured using a ×10 objective lens (1024 × 1024 pixels). The number of coronary vessels were counted manually with ImageJ. To quantify coronary vessel expansion in regenerating hearts from *deltaC:EGFP* fish: three medial, longitudinal sections were selected from each heart. Images of single optical slices of the ventricle were acquired using a ×20 objective lens (1024 × 1024 pixels) by adjusting the gain to detect EGFP signals above background level. EGFP⁺ areas were quantified in pixels by ImageJ software, and the density of EGFP⁺ signals versus imaged area (500 × 300 pixels) was calculated and averaged from 3 sections of each ventricle for each heart. To quantify coronary vessel sprouts associated with *hapln1a*⁺ cells, whole-mounted specimens were selected and imaged. Images of ventricles were captured using a ×20 objective lens (1024 × 1024 pixels). The correlation between coronary vessel sprouts and *hapln1a*⁺ cells was determined by the percentage of coronary vessel sprouts associated with *hapln1a*⁺ cells in the following orientations: pre-leading, in parallel with, or behind the coronary vessel sprouts. To quantify the association of coronary vessel sprouts with *serpine1* signals in juvenile

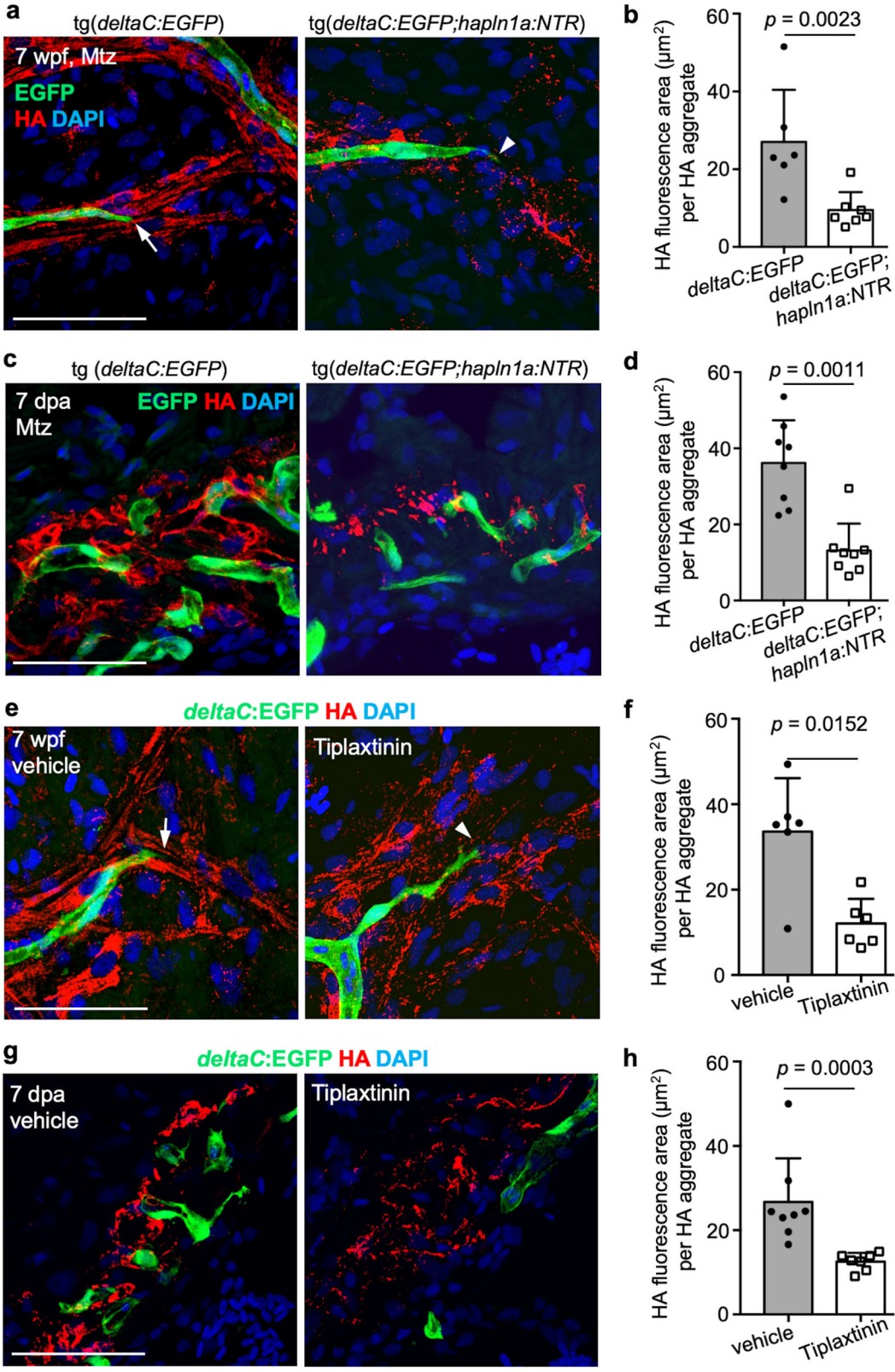

ventricles, whole-mounted specimens were selected and imaged. Images of ventricles were captured using a ×20 objective lens (1024 × 1024 pixels). The correlation between coronary vessel sprouts and *sepine1* signals was determined by the percentage of coronary vessel sprouts associating with *serpine1* within 30 μm of the coronary vessel sprouts.

## HA quantification

To calculate HA signals, whole-mount ventricles in juvenile fish and ventricular sections of injured hearts from adult fish were stained with biotinylated HABP (hyaluronan binding protein) and fluorescently labeled streptavidin–Alexa-647 conjugate. Images of the ventricle were captured with a ×20 objective lens (1024 × 1024 pixels). HA signals

**Fig. 9 | Depletion of hapln1a+ cells or serpine1 pathway inactivation lead to defective hyaluronic acid (HA) deposition in the coronary growth area during morphogenesis and regeneration. a** Whole-mount views of coronary cells and HA signals on the ventricular surface of juvenile *deltaC:EGFP* fish, treated with Mtz for 2 days and hearts collected at 5 days post treatment. Arrows represent the coronary sprouts lining with organized HA signals and arrowheads represent the coronary sprouts without organized HA signals. Scale bars, 50 μm. **b** Quantification of HA⁺ signal area per HA aggregate from experiments in **a**. $n = 6$ deltaC:EGFP and $n = 7$ deltaC:EGFP;hapln1a:NTR fish were used. Mann–Whitney rank-sum test (two-sided). Data are presented as mean values ± SD. Source data are provided as a Source Data file. **c** Section images of ventricles of *deltaC:EGFP* and *deltaC:EGFP;hapln1a:NTR* fish treated with Mtz from 2 dpa for 3 continuous days and hearts were collected at 7 dpa and assessed for HA signals in the injury site. Scale bar, 50 μm. **d** Quantification of the HA⁺ signal area per HA aggregate in the injury edges from experiments in **c**. 8 fish for each group were analyzed. Mann–Whitney rank-sum test (two-sided). Data are presented as mean values ± SD. Source data are

provided as a Source Data file. The experiment was repeated once with similar results. **e** Whole-mount views of coronary cells and HA signals on the ventricular surface of juvenile *deltaC:EGFP* fish, treated with vehicle and Tiplaxtinin for 7 continuous days. Arrows represent the coronary growth extensions lining with organized HA signals and arrowheads represent the coronary sprouts without organized HA signals. Scale bars, 50 μm. **f** Quantification of HA⁺ signal area per HA aggregate from experiments in **e**. 6 animals were used for each group. Mann–Whitney Rank Sum test (Two-sided test). Data are presented as mean values ± SD. Source data are provided as a Source Data file. The experiment was repeated once with similar results. **g** Section images of ventricles of *deltaC:EGFP* fish treated with vehicle or Tiplaxtinin from 3 to 6 dpa, and analyzed at 7 dpa, assessed for HA signals in the injury site. Scale bar, 50 μm. **h** Quantification of the HA⁺ signal area per HA aggregate in the injury edges from experiments in **g**. $n = 8$ vehicle control and $n = 7$ inhibitor-treated fish were used. Mann–Whitney rank-sum test (two-sided). Data are presented as mean values ± SD. Source data are provided as a Source Data file. The experiment was repeated once with similar results.

were measured in pixels by ImageJ software for signals either close to vessel tips (200 × 200 pixels) of whole-mount ventricles or in the edge of the injury site (250 × 250 pixels) of each sectioned ventricle. We calculated HA aggregates by the number of HA dots per unit area of HA signals to determine the HA structure organization.

## Single-cell RNA-sequencing

To prepare *tcf21⁺* and *hapln1a⁺* cells for single-cell RNA-sequencing analyses: 80 *tcf21:nucEGFP* and 120 *hapln1a:EGFP* fish were raised to the juvenile stage, and 60 *hapln1a:EGFP* fish were raised to the adult stage. Juvenile hearts were collected at 7 wpf (1.6–1.8 cm length). The hearts of adult fish at 6 months old (3.5–4 cm length) were extracted at 7 days post-amputation (dpa) and the wounded apex region was collected. The heart samples were digested with 0.26 U/mL Liberase™ Thermolysin Medium (TM) based on a previously published protocol[102]. Dissociated cells were spun down and live EGFP⁺ cells were sorted by flow cytometry with FACSDiva v9.0. To ensure at least a 95% cell viability following the entire procedure, we gated viable cells by negative SYTOX™ Red fluorescence and examined cell viability after cell sorting (Supplementary Fig. 1). Isolated cells were sent to the Emory Integrated Genomics Core (EIGC) center for 10x single-cell RNA-sequencing. Isolated EGFP⁺ cells were prepared in a single-cell suspension and counted using a Countess (ThermoFisher) system. The loaded Single Cell 3′ Chip was placed on a 10x Genomics Chromium Controller Instrument (10x Genomics, Pleasanton, CA, USA) to generate single-cell gel beads in emulsion (GEMs). Single-cell RNA-seq libraries were prepared using the Chromium Single Cell 3′ Library & Cell Bead Kit v3.1 (Cat. No. 1000128, 1000127, 120262; 10x Genomics) according to the manufacturer's protocol. Libraries were sequenced with an Illumina NextSeq550 using mid-output 150-cycle kits according to manufacturer specifications. The newly generated scRNA-seq data were demultiplexed, aligned, and quantified using Cell Ranger Single-Cell Software V3.0.2. Preliminary filtered data generated from Cell Ranger were used for downstream analysis by the Seurat R package according to standard workflow.

## Statistics and reproducibility

All data are presented as mean ± standard deviation. All statistical analyses were performed using GraphPad Prism 7 software. The two-sided Mann–Whitney rank-sum test was used for assessing statistical differences between two groups. Animals in experimental groups were randomly assigned and blinded for quantification. The results with *P* values < 0.05 were considered statistically significant.

## Reporting summary

Further information on research design is available in the Nature Portfolio Reporting Summary linked to this article.

## Data availability

The scRNA-seq dataset has been deposited in the National Center for Biotechnology Information Gene Expression Omnibus (GEO) under accession number GSE216649. All other relevant data supporting the key findings of this study are available within the article and its Supplementary Information files or from the corresponding author upon reasonable request. Source data are provided with this paper.

## Code availability

Code used for single-cell RNA-seq data analysis is available on a GitHub repository at https://github.com/jishengsun/tcf21_hapln1a_scRNAseq.

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

## Acknowledgements

We thank Weilan Lin for fish care and the Developmental Studies Hybridoma Bank for antibodies. Microscopy data for this study were acquired and analyzed using the Microscopy in Medicine Core in Cardiology at Emory. This work was supported by a NIH T32 training fellowship (5T32HL007745-28) and by a AHA (23POST1014396) to E.A.P.; grants from NHLBI (R01HL142762) to J.W. The Microscopy in Medicine Core in Cardiology at Emory was supported by NIH grant (P01 HL095070).

## Author contributions

Conceptualization by J.S., and J.W.; methodology by J.S., and J.W.; investigation by J.S., E.A.P., X.C., and J.W.; formal analysis by J.S., and J.W.; visualization by J.S. and J.W.; writing by J.W.; funding acquisition by J.W; supervision by J.W. All authors commented on the manuscript.

## Competing interests

The authors declare no competing interests.
