## [Peer Review File · Nature Communications]

REVIEWER COMMENTS

Reviewer #1 (Remarks to the Author):

This paper describes a close correlation between growing coronary vessels during zebrafish heart development and regeneration with a subpopulation of epicardial cells that are positive for hapln1a. In a series of convincing, high quality imaging in vivo and ex vivo the authors in particular show that hapln1a+ cells appear to guide coronary vessel growth. Genetic ablation of hapln1 shows that these cells are indeed important for coronary vessel growth. Several single cell sequencing profiles are presented (for epicardial cells and hapln1a+ cells) which are a nice resource for further studies.

While these results are very interesting, since they add to the author's previous findings that the hapln1a+ epicardium is also important for other events during heart regeneration, in particular for cardiomyocyte proliferation, I find the molecular mechanistic insight into how hapln1a+ epicardial cells regulate coronary vessel growth not substantial enough for Nat. Comm. While the authors present serpin1 as a candidate factor produced by the hapln1a+ cells that guides vessel growth, this hinges on drug inhibitor experiments only. As far as I can tell, whether the drug actually inhibits zebrafish serpin and how specific it is, has not been shown. Thus, additional evidence, ideally by genetic means should be provided. I'm aware that this is technically very challenging (e.g. if non-conditional mutants are embryonic lethal), but nevertheless think that the paper in its current form is somewhat too light on molecular mechanisms for Nat. Comm.

Additional major issues:

1) since development and growth of fish is heavily influenced by environmental factors, all stage descriptions should not be given in time after fertilization, but in the "standardized standard length" scheme established by David Parichy. <https://www.ncbi.nlm.nih.gov/pmc/articles/PMC3030279/>

2) Methods: transgenic lines must be identified by their unique allele designation given by the Zebrafish nomenclature committee (ZFIN)

3) Methods are generally too sparse and won't allow repetition of experiments since essential information is missing. E.g. the description of the FACS for single-cells preparation should include information on how debris / dead cells were excluded, gates should be published in the supplement.

4) Where is the data supporting this statement? "...notable difference in cluster 3 with high levels of cell proliferation related genes".

4) Line 204: “The expression of *ciarta* and *dbpb* in cluster 5 cells suggests an involvement in the rhythmic process”. Please elaborate; which “rhythmic process”?

5) “we observed significant *hapln1a* expression in clusters 1 and 3”. Fig. 1D shows *hapln1a* also in clusters 4 and 5; what’s the basis for ignoring these and for focusing on 1 and 3?

6) Fig. 2F: text and figure legends mention “AcT-Hu+” cells, methods mention anti-acetylated tubulin and anti-HuC/HuD antibodies. 2F shows the “AcT-Hu+” cells in the red channel. Authors must clarify how this was done. Were hearts co-stained with both antibodies, which were both detected in the red channel? How do authors then know these cells are double positive for acT and Hu?

Reviewer #2 (Remarks to the Author):

In this manuscript, Sun et al. describe a role for *hapln1a*⁺ cells for guiding coronary artery growth during juvenile heart development and adult heart regeneration in the zebrafish model system. They make use of nice reporter and conditional cell ablation tools to show that *hapln1a*⁺ cells of the epicardium are associated as “shear structures” at the tip ahead of coronary endothelium, in contrast to macrophages, neurons, or cardiomyocytes. Using extracted cultured hearts they provide evidence for tracking of vessels along a *hapln1a* pathway, and that following ablation of *hapln1a*⁺ cells (via NTR-MTZ killing) the vessel growth fails. Similar results are shown for regenerating adult hearts. *Serpine1* is identified as a gene expressed in the *hapln1a*⁺ cells, and a chemical blocker of *Serpine1* also blocked artery growth and regeneration. The manuscript is well written and the topic is important.

Major Issues.

1) The point of scRNA-seq described in Fig. 1 is unclear. The authors published the same analysis previously earlier this year, although somehow a different number of clusters were identified. Other scRNA-seq analyses have been published by the Poss and Cao groups using presumably the exact same cell populations (*tcf21*⁺ epicardium). Why are different clusters being shown than was shown previously and for what purpose? More importantly, the data shown does not meet the description. For example, the authors claim enrichment of *hapln1a* cells in clusters 1 and 3. There is no obvious enrichment in cluster 3. Violin plots were done for many genes but strangely not for *hapln1a*, but anyway it is unclear why this data is shown and what value it adds.

2) The major issue by far is the lack of evidence that there is anything functionally specific about the hapln1a+ cells. The data showing association with tips is not very compelling in the images shown, and the authors could do a more rigorous job to quantify this.

3) More importantly, no experiments are carried out to show that there is anything special about the hapln1+ cells beyond an association. The ablation of tcf21+ cells has been shown to impede coronary vessel growth and regeneration in previous publications. Here the authors show similar effects with ablation of hapln1a+ cells. However, the hapln1a+ cells comprise a significant part of the epicardium (33% in homeostasis, 50% in regenerating hearts??). The proper control is to ablate a similar proportion of hapln1a- cells, to determine if the role in coronary growth is really specific to the hapln1a+ population. This is not a small ask, but is really essential for interpreting the results especially for a journal with impact of Nature Comm. At the very least, one could compare a partial 50% random ablation using tcf21:NTR with the full ablation of hapln1a:NTR. Does the latter show a much stronger phenotype?

4) Details for inhibition of Serpine1 were missing. Is the concentration used equivalent to just what is needed to inhibit activity? Can inhibition of activity be measured? A complementary genetic approach would be highly desirable to exclude chemical off-targets.

Minor issues

1) The term “shear structures” needs explanation as it is not a commonly used descriptor. Is there some evidence for causation of shear stresses?

2) What does it mean that vessel number is increased by hapln1a cell ablations? Are these different vessels or fragments of previous vessels?

Reviewer #3 (Remarks to the Author):

In this study Sun et al identify a subset of epicardial cells that express Hapln1a which are associated with sprouting coronary vessels in the juvenile and regenerating zebrafish hearts, and use ablation experiments to show that Hapln1a is required for coronary growth in both scenarios. They further

identify that *serpine1*, which encodes for a negative regulator of plasminogen activator, is expressed in *hapln1a*⁺ epicardial cells, is found at coronary tips, and is required for vascularisation both in coronary development and regeneration.

Previous studies have identified that epicardial subtypes exist, that the epicardium supports coronary development, that *hapln1a* is expressed in the epicardium, that *hapln1a* is required for heart regeneration, and that *serpine1* also promotes heart regeneration. However, little is understood about heterogeneity in the epicardium, how the epicardium supports coronary growth, and which epicardial cells are required. In this context, the study by Sun et al contributes new knowledge to the field of coronary development, and unifies and extends previous studies on the role of epicardium in vascularisation and revascularisation in development and injury. The experiments are well designed and executed, and the data is well presented. However, the depth of mechanistic analysis of the roles of *hapln1a* and *serpine1* in vascularisation, particularly in terms of their functions and relationship, is a little unsatisfactory. What is the impact of *Hapln1a* and *Serpine1* LOF on the ECM around the vasculature for example? How/why do these ECM modifications support vessel formation? And what is the relationship between them? While further experiments would help to provide better insights into the specific role of the *Hapln1a*⁺ *Serpine1*⁺ epicardium in supporting coronary (re)development, the authors also do not really dive into these questions in the discussion, which is an omission. Taking steps to address the mechanism would significantly improve the impact of the manuscript.

Major comments:

Relating to the comment above, the authors do not investigate changes in ECM in either the *Hapln1a* or *Serpine1* models, or discuss this mechanism of action of each gene. The authors could assess for example HA presence/organisation (as in their previous Circulation *Hapln1a* epicardium paper) around the vessels in the juvenile stage in wild type and upon ablation of *Hapln1a*⁺ cells or Tiplaxtinin treatment. In Line 339 – the authors also suggest that *hapln1a* regulates coronary vessel development through *Serpine1* and that they will test this – of course this will not be happening at a transcriptional level, and since the authors also don't look in detail at the mechanisms of action of either protein, or the impact on the extracellular environment (showing only that they both promote coronary development), the experiments that they perform subsequently don't really address this hypothesis.

It appears that not all vessel tips are associated with *Hapln1a* (or even *Serpine1*) for example the 6 hour timepoint shown in Figure 2 – there appear to be sprouts forming without any *Hapln1a*. Related to this, some revascularisation still occurs in the *hapln1a* KO. Does this mean *Hapln1a* is only required by a subset of branching vessels? To investigate this, the authors should quantify the number of tips with/without *Hapln1a* and *Serpine1*, and comment on this possible heterogeneity.

There are some assertions or observations in the paper that could be better supported through the extensive single cell sequencing data already included in the paper. For example:

The authors state that the location of hapln1a cells indicates they function as perivascular cells (Line 221) – are there perivascular markers in the hapln1a+ clusters? If so, which cluster?

The authors also identify 3 different kinds of hapln1a+ cell – 2 of which behave more like perivascular cells, surrounding the vessels, and 1 of which (the most prevalent kind, which ‘precedes’ the growing sprout and is Serpine1 + in juveniles) exhibits behaviour unexpected of a perivascular cell. Do the authors think these cell types are from different Hapln1+ clusters, or are they all the same subtype? In the single cell seq data presented in figure S2 (Hapln1+ sorted), serpine1 is found in a relatively small proportion of those cells – do these form a distinct cluster separate from other Hapln1a+ cells? Does clustering of the Hapln1a+ sc-RNA-seq data give more insights into this population?

In Fig S2, serpine1 is expressed at very low levels in Hapln1a+ cells of juvenile hearts compared to regenerating hearts – the authors should comment on this difference. Related to this, the phenotype in the Serpine1-inhibited hearts at 7wpf (Fig. 7C) is very profound given the small number of cells that express Serpine1 (Fig S2). In the explanted hearts (Fig 7A), there even appears to be a reduction in vessels – is this the impact not of defective growth, but rather death/regression of the existing vascular network? The authors should comment on these issues.

Minor comments:

The authors state there is significant enrichment of Hapln1a in the angiogenic epicardial subclusters of 1 and 3 (Line 208). In Figure 1 it is clear there is also significant expression in clusters 4 and 5, and so this statement is a little misleading (arguably, cluster 4 actually has more hapln1a+ expressing cells than cluster 3).

Did the authors confirm that the Tiplaxtinin treatment is blocking serpine1 activity/function? For example by assaying conversion of plasminogen to plasmin.

Figure 1 – there is a lack of consistency in the colour coding used in the heatmaps - sometimes blue represents the highest value, sometimes the lowest – this could be consistent within the figure to simplify.

Figure 4 – The authors show that the vessel density and junction number (graphs D and E) are reduced, which is in line with the observation that vascular growth is less in the hapln1a ablation model. However, vessel number per area is increased (F), which seems contradictory to the previous quantifications. Can the authors explain the difference between these two quantifications?

Figure 4 and 5 – the graphs are distorted along the x axis – aspect ratio should be maintained.

Figure 6 – The *serpine1* expression shown in the images doesn't seem to colocalise with Hapln1a at terminal branches (rather it is localised more centrally within the Hapln1a network) – is *Serpine1* expressed broadly within the Hapln1a network as well as around tip cells?

Below is a point-by-point response to the reviewers' comments.

Reviewer 1

This paper describes a close correlation between growing coronary vessels during zebrafish heart development and regeneration with a subpopulation of epicardial cells that are positive for *hapln1a*. In a series of convincing, high quality imaging *in vivo* and *ex vivo* the authors in particular show that *hapln1a*⁺ cells appear to guide coronary vessel growth. Genetic ablation of *hapln1* shows that these cells are indeed important for coronary vessel growth. Several single cell sequencing profiles are presented (for epicardial cells and *hapln1a*⁺ cells) which are a nice resource for further studies. While these results are very interesting, since they add to the author's previous findings that the *hapln1a*⁺ epicardium is also important for other events during heart regeneration, in particular for cardiomyocyte proliferation, I find the molecular mechanistic insight into how *hapln1a*⁺ epicardial cells regulate coronary vessel growth not substantial enough for Nat. Comm.

A: We appreciate the reviewer's comments and per the reviewer's suggestions we have performed additional experiments to further analyze the effect of *hapln1a*⁺ cells and *serpine1* on the ECM component hyaluronan (HA) for further understanding of the underlying mechanism. With our new analyses, we observed HA lining coronary vessel extensions and also preceding these vessels. Further, we discovered that depletion of *hapln1a*⁺ cells disrupted the HA organization and inhibiting *serpine1* activity led to a disorganized HA structure. Our new experiments indicate that HA deposition by *hapln1a*⁺ cells and then organization by *serpine1*-expressing cells is required for coronary growth during morphogenesis and regeneration. Although previous tissue engineering studies have demonstrated that HA-based materials are sufficient to stimulate angiogenesis and vascular growth and HA is required for zebrafish coronary re-vascularization (Brandes et al., JCI 1991; Peattie et al., Europe PMC 2004; Turner et al., Biomaterials 2004; Lepidi et al., Eur J Vasc Endovasc Surg 2006; Pardue et al., Organogenesis 2008; Burdick and Prestwich, Adv Mater 2011; Munch et al., Development 2017; Ghose et al., BMC Cancer 2018; Luo et al., Front Bioeng Biotechnol 2022), there is no direct observation of organized HA structures along and preceding coronary growth *in vivo*. Further, no reports exist regarding the regulation mechanism of HA by *hapln1a*⁺ cells and *serpine1*. We have updated the manuscript to include these new results on pages 14-16, and in Fig. 9 of the revised manuscript:

"To elucidate the mechanism by which *hapln1a*⁺ cells and *serpine1* activity impact coronary growth, we assessed the distribution of the *hapln1a*⁺ cell substrate, hyaluronic acid (HA). We recently reported that *hapln1a*⁺ epicardial cells regulate HA deposition to facilitate cardiomyocyte proliferation during heart morphogenesis and regeneration [48]. HA is required for coronary revascularization during zebrafish heart regeneration [79-81]. Previous studies also demonstrated that endothelial cells can attach and grow along HA fibers *in vitro*, and a scaffold in tissue-engineered vascular grafts formed from HA-based biomaterials can completely generate a new vascular tube. These studies indicate that the administration of HA can enhance cell proliferation, adhesion, tubular sprout formation, and the migration of endothelial cells [79-85]. We speculate that HA is involved in guided coronary growth. To examine this possibility, we first assessed HA localization with coronary vessels on the ventricular surface in juvenile *deltaC:EGFP* animals. We

detected strong association of linear HA signals with EGFP⁺ signals (Fig. 9a). Next, we examined the HA deposition after ablating *hapln1a*⁺ cells at 7 wpf. Juvenile *hapln1a:mCherry-NTR;deltaC:EGFP* animals and *deltaC:EGFP* siblings were treated with 10 mM Mtz and their hearts were collected for histological analysis. We assessed the size of HA aggregates by quantifying the area of each focus of HA and found that *hapln1a*⁺ cell-depleted hearts displayed 35% lower HA intensity per HA aggregate, when compared with control siblings (Fig. 9b), indicating that the ablation of *hapln1a*⁺ cells disrupted the linear structure of HA in the coronary growth area. Next, we examined HA signals in the regeneration area and found HA signals were closely associated with *deltaC:EGFP*⁺ cells (Fig. 9c). We then assessed the HA deposition within the regenerating area after depleting *hapln1a*⁺ cells and observed that 7 dpa injury sites of *hapln1a*⁺ cell-depleted animals displayed ~47% smaller puncta of HA, compared with wild-type siblings (Fig. 9d). Next, we assessed whether *serpine1* functions through regulating the HA structure during morphogenesis and regeneration, as previous reports indicated *serpine1* protects the ECM to maintain a matrix necessary for endothelial cells to migrate and form capillaries [86]. We examined HA deposition in the ventricle surface after treating juvenile *deltaC:EGFP* fish with Tiplaxtinin for 12 hours and continuously for 7 days and found that inhibitor treated hearts displayed a 64% lower HA intensity per HA aggregate, when compared with control siblings (Fig. 9e and f), revealing a disorganized HA linear structure within the coronary growth area. Lastly, we treated adult fish with Tiplaxtinin from 3 dpa for 12 hours and continuously for 4 days and observed 53% lower HA intensity per HA aggregate at 7 dpa, compared with vehicle controls (Fig. 9g and h). Our results indicate a mechanism for *hapln1a*⁺ cell function during morphogenesis and regeneration, in which HA is deposited by *hapln1a*⁺ cells and then organized by *serpine1*-expressing cells for coronary growth during vascularization and revascularization.”

While the authors present *serpin1* as a candidate factor produced by the *hapln1a*⁺ cells that guides vessel growth, this hinges on drug inhibitor experiments only. As far as I can tell, whether the drug actually inhibits zebrafish *serpin* and how specific it is, has not been shown. Thus, additional evidence, ideally by genetic means should be provided. I'm aware that this is technically very challenging (e.g. if non-conditional mutants are embryonic lethal), but nevertheless think that the paper in its current form is somewhat too light on molecular mechanisms for Nat. Comm.

A: We agree with the reviewer that a genetic approach is an excellent means to exclude possible chemical off-target effects and drug non-specific effects/efficiency. As the reviewer mentioned, genetic means of manipulating adult zebrafish genes are time-consuming and challenging/risky: 1) more than 1 year to develop a CRISPR/Cas9 *serpine1* mutant generation and several generation outcrosses; and 2) a *serpine1* mutant also does not guarantee that there will be no compensation effects. For these reasons, we chose a pharmacological inhibition approach to perform functional analysis of *serpine1* during heart regeneration. We chose the drug Tiplaxtinin as its *serpine1* inhibition activity has been previously measured and was utilized for the pharmacological inhibition of *serpine1* activity and published by different research groups (Gorlatova et al., JBC 2007; Daniel et al., PLoS One 2015; Kang et al., Oncotarget 2016; Munch et al., Development 2017; Lin et al., Cell Rep. 2020; Yamagami et al., BBRC 2020). Natalia V. Gorlatova demonstrated the capacity for Tiplaxtinin to block *serpine1* activity, and IC₅₀

values for *serpine1* inactivation by Tiplaxtinin are from 9 to 12 μM . The inhibitory effect of 20 μM Tiplaxtinin for *serpine1*, the concentration which we used in this MS, is about 95%. As the means to examine possible downstream effectors of *serpine1*, like urokinase plasminogen activator (uPA) and tissue plasminogen activator (tPA) is currently still lacking in zebrafish, we are unable to examine the effect of *serpine1* inhibition. However, we performed experiments to exclude the possible toxic effects of Tiplaxtinin by examining coronary growth after removing the drug. We found that inhibited coronary growth restarted after removing the inhibitor. We have now included these data on pages 13-14, and in Supplementary Fig. 8 of the revised manuscript.

“...by treatment with its antagonist Tiplaxtinin, which blocks Serpine1 protease and has been used to inhibit *serpine1* activity in different systems, including during zebrafish heart regeneration [70, 73-77]. Previous studies measured and determined that the inhibitory effect of 20 μM of Tiplaxtinin on *serpine1* is around 95% [73, 78].”

“Moreover, these blocked coronary extensions can re-start after drug removal, indicating that the coronary growth restriction by *serpine1* inhibition does not occur by coronary cell death (Supplementary Fig. 8).”

Additional major issues:

1) since development and growth of fish is heavily influenced by environmental factors, all stage descriptions should not be given in time after fertilization, but in the “standardized standard length” scheme established by David Parichy.

<https://www.ncbi.nlm.nih.gov/pmc/articles/PMC3030279/>

A: We have now added the length information of fish we used at both juvenile and adult stages in the revised manuscript, on page 19, paragraph 2:

“... transgenic zebrafish were used at the age of 6-7 weeks with a length between 1.5 – 1.8 cm (juvenile), or 4-12 months with a length of at least 3 cm (adult).”

and page 24, paragraph 1.

“Juvenile hearts were collected at 7 wpf (1.6 – 1.8 cm length). The hearts of adult fish at 6 months old (3.5 – 4 cm length)...”

2) Methods: transgenic lines must be identified by their unique allele designation given by the Zebrafish nomenclature committee (ZFIN)

A: The information for all transgenic lines according to their unique allele designation given by the ZFIN are now included in the revised manuscript, on page 19, paragraph 2:

“Transgenic strains described elsewhere include: $\text{Tg}(gata4:EGFP)^{ae1\text{Tg}}$ and $\text{Tg}(tcf21:nucEGFP)^{pd41\text{Tg}}$ [100]; $\text{Tg}(hapln1a:EGFP)^{pd338\text{Tg}}$ and $\text{Tg}(hapln1a:mCherry-NTR)^{em14\text{Tg}}$ [48]; and $\text{Tg}(\delta C:EGFP)^{em11\text{Tg}}$ and $\text{Tg}(\delta C:mCherry)^{em11\text{Tg}}$ [47].”

3) Methods are generally too sparse and won't allow repetition of experiments since essential information is missing. E.g. the description of the FACS for single-cells preparation should include information on how debris/dead cells were excluded, gates should be published in the supplement.

A: The updated manuscript now provides a revised but concise description in the methods section that is sufficient for experimental repetition. The information for FACS for scRNA-seq is now included in the Methods and supplementary data. The updated Methods information is now available on pages 19-24, and Supplementary Fig. 1 in the revised manuscript.

4) Where is the data supporting this statement? "...notable difference in cluster 3 with high levels of cell proliferation related genes".

A: In the revised manuscript, we have now included feature plots for *frzb*, *mustn1a* and *tgfb1* expression in *tcf21⁺* clusters of the hearts at 7 wpf. These genes were previously indicated to be associated with the cell proliferation process. We have included this information in page 6, paragraph 1, and in Supplementary Fig. 2 in the revised manuscript.

"...we observed a notable difference in cluster 3 with high levels of expression of genes related to cell proliferation, such as *frzb*, *mustn1a*, and *tgfb1* [31-33]."

4) Line 204: "The expression of *ciarta* and *dbpb* in cluster 5 cells suggests an involvement in the rhythmic process". Please elaborate; which "rhythmic process"?

A: Both *ciarta* and *dbpb* are suggested to have a role in the circadian rhythm. This information is now included in the Results section on page 6, paragraph 1:

"The expression of *ciarta* and *dbpb* in cluster 5 suggests an involvement in the circadian rhythm process [38, 39]."

5) "we observed significant *hapln1a* expression in clusters 1 and 3". Fig. 1D shows *hapln1a* also in clusters 4 and 5; what's the basis for ignoring these and for focusing on 1 and 3?

A: We performed Violin plot analysis comparing the expression of *hapln1a* in *tcf21⁺* cell clusters. *hapln1a* shows high expression in cluster 1, slight expression in clusters 3, 4, 5, and little expression occurs in clusters 2 and 6. We have now revised the description on page 7, paragraph 1:

"As cluster 1 is the most heavily represented of these 3 cell states, we first focused on cluster 1 and observed high expression of *hapln1a* in this cluster [23], with slight expression in clusters 3, 4 and 5 (Fig. 1d and Supplementary Fig. 3)."

6) Fig. 2F: text and figure legends mention "AcT-Hu+" cells, methods mention anti-acetylated tubulin and anti-HuC/HuD antibodies. 2F shows the "AcT-Hu+" cells in the red channel. Authors must clarify how this was done. Were hearts co-stained with both antibodies, which were both detected in the red channel? How do authors then know these cells are double positive for acT and Hu?

A: We apologize for the confusion and have provided an explanation in the revised manuscript and here. This assay used a combination of antibodies against AcT and HuC/D to identify intracardiac neuronal somata and axons. The utility of this label combination to detect neuronal somata has been established in previous neuroanatomical studies such as in the zebrafish intestine (Bisgrove et al., J Neurobiol 1997; Olsson, Acta Histochemica 2009) and in the goldfish heart (Newton et al., JCN 2014). In order to label all neuronal somata, for regional and total counts of neurons, we

employed this AcT-Hu immunohistochemistry in this study. With this method, both AcT and Hu primary antibodies are recognized through the utilization of an Alexa-Fluor 594 goat anti-mouse secondary antibody and then visualized together with the Zeiss LSM800 confocal microscope. As we sought to examine the possible relationship of coronary vessels with all nerves, we only examined the nerves as combined AcT+ and Hu+ signals instead of attempting to examine AcT+ or Hu+ signals in isolation. We included the description in page 9, paragraph 1:

“... an established label combination which has previously been used to detect neuronal somas in neuroanatomical studies in the zebrafish intestine and goldfish heart [60-62].”

Reviewer 2

In this manuscript, Sun et al. describe a role for hapln1a+ cells for guiding coronary artery growth during juvenile heart development and adult heart regeneration in the zebrafish model system. They make use of nice reporter and conditional cell ablation tools to show that hapln1a+ cells of the epicardium are associated as “shear structures” at the tip ahead of coronary endothelium, in contrast to macrophages, neurons, or cardiomyocytes. Using extracted cultured hearts they provide evidence for tracking of vessels along a hapln1a pathway, and that following ablation of hapln1a+ cells (via NTR-MTZ killing) the vessel growth fails. Similar results are shown for regenerating adult hearts. Serpine1 is identified as a gene expressed in the hapln1a+ cells, and a chemical blocker of Serpine1 also blocked artery growth and regeneration. The manuscript is well written and the topic is important.

Major Issues.

1) The point of scRNA-seq described in Fig. 1 is unclear. The authors published the same analysis previously earlier this year, although somehow a different number of clusters were identified. Other scRNA-seq analyses have been published by the Poss and Cao groups using presumably the exact same cell populations (*tcf21*+ epicardium). Why are different clusters being shown than was shown previously and for what purpose?

A: We apologize for the confusion and provide an explanation here, as well as in the revised manuscript. Previously, the scRNA-seq analyses of the epicardium from the Poss Lab, Cao Lab, Riley Lab, and our lab (Wang Lab) were performed with the epicardium of the embryonic heart, adult uninjured heart, or adult regenerating hearts. The scRNA-seq analysis performed in this manuscript utilized the juvenile zebrafish heart, a developmental stage at which scRNA-seq of the epicardium has not been reported. As cardiac regeneration recapitulates many aspects of heart morphogenesis and critical biological events like coronary vascularization and myocardial compaction occur during this juvenile stage, our scRNA-seq analysis provided in this manuscript will be informative to thoroughly understanding epicardial function at different life stages and during the regeneration process. The information regarding the use of juvenile zebrafish hearts for *tcf21*+ scRNA-seq is provided on page 6, paragraph 1 of revised manuscript:

“Zebrafish coronary growth establishes a dense vasculature network from 5 to 6 weeks post-fertilization (wpf) until the adult stage, a period during which latent epicardial clusters are also undergoing development [23]. As cardiac regeneration recapitulates many aspects of heart morphogenesis and detection of epicardial cell clusters at the juvenile stage has not been reported, we performed scRNA-seq analysis of *tcf21*+ cells [25] in zebrafish hearts at 7 wpf (Supplementary Fig. 1) to explore juvenile epicardial clusters and their potential effects on coronary growth.”

More importantly, the data shown does not meet the description. For example, the authors claim enrichment of hapln1a cells in clusters 1 and 3. There is no obvious enrichment in cluster 3. Violin plots were done for many genes but strangely not for hapln1a, but anyway it is unclear why this data is shown and what value it adds.

A: We carefully checked our data and the corresponding description in the revised manuscript to make sure the data shown meet the description. We agree with the reviewer

that the manuscript requires a Violin plot for *hapln1a*. We performed Violin plot analysis for *hapln1a* and observed high expression of *hapln1a* in cluster 1, slight expression in clusters 3, 4, 5, and little expression in clusters 2 and 6. We have revised the description on page 7, paragraph 1 of the updated manuscript:

“As cluster 1 is the most heavily represented of these 3 cell states, we first focused on cluster 1 and observed high expression of *hapln1a* in this cluster [23], with slight expression in clusters 3, 4 and 5 (Fig. 1d and Supplementary Fig. 3).”

2) The major issue by far is the lack of evidence that there is anything functionally specific about the *hapln1a*⁺ cells. The data showing association with tips is not very compelling in the images shown, and the authors could do a more rigorous job to quantify this.

A: We appreciate the reviewer’s comments and per the reviewer’s suggestions we have performed additional experiments to further analyze the effect of *hapln1a*⁺ cells and *serpine1* on the ECM component hyaluronan (HA) for further understanding of the underlying mechanism. With our new analyses, we observed HA lining coronary vessel extensions and also preceding these vessels. Further, we discovered that depletion of *hapln1a*⁺ cells disrupted the HA organization and inhibiting *serpine1* activity led to a disorganized HA structure. Our new experiments indicate that HA deposition by *hapln1a*⁺ cells and then organization by *serpine1*-expressing cells is required for coronary growth during morphogenesis and regeneration. Although previous tissue engineering studies have demonstrated that HA-based materials are sufficient to stimulate angiogenesis and vascular growth and HA is required for zebrafish coronary re-vascularization (Brandes et al., JCI 1991; Peattie et al., Europe PMC 2004; Turner et al., Biomaterials 2004; Lepidi et al., Eur J Vasc Endovasc Surg 2006; Pardue et al., Organogenesis 2008; Burdick and Prestwich, Adv Mater 2011; Munch et al., Development 2017; Ghose et al., BMC Cancer 2018; Luo et al., Front Bioeng Biotechnol 2022), there is no direct observation of organized HA structures along and preceding coronary growth *in vivo*. Further, no reports exist regarding the regulation mechanism of HA by *hapln1a*⁺ cells and *serpine1*. We also conducted a more rigorous analysis on the position of *hapln1a*⁺ cells and coronary vessel sprouts and these additional quantifications are now included in Fig. 2d of the revised manuscript. We have updated the manuscript to include these new experiments on pages 14-16, and in Fig. 9 of the revised manuscript:

“To elucidate the mechanism by which *hapln1a*⁺ cells and *serpine1* activity impact coronary growth, we assessed the distribution of the *hapln1a*⁺ cell substrate, hyaluronic acid (HA). We recently reported that *hapln1a*⁺ epicardial cells regulate HA deposition to facilitate cardiomyocyte proliferation during heart morphogenesis and regeneration [48]. HA is required for coronary revascularization during zebrafish heart regeneration [79-81]. Previous studies also demonstrated that endothelial cells can attach and grow along HA fibers *in vitro*, and a scaffold in tissue-engineered vascular grafts formed from HA-based biomaterials can completely generate a new vascular tube. These studies indicate that the administration of HA can enhance cell proliferation, adhesion, tubular sprout formation, and the migration of endothelial cells [79-85]. We speculate that HA is involved in guided coronary growth. To examine this possibility, we first assessed HA localization with coronary vessels on the ventricular surface in juvenile *deltaC:EGFP* animals. We detected strong association of linear HA signals with EGFP⁺ signals (Fig. 9a). Next, we examined the HA deposition after ablating *hapln1a*⁺ cells at 7 wpf. Juvenile

hapln1a:mCherry-NTR;deltaC:EGFP animals and *deltaC:EGFP* siblings were treated with 10 mM Mtz and their hearts were collected for histological analysis. We assessed the size of HA aggregates by quantifying the area of each focus of HA and found that *hapln1a*⁺ cell-depleted hearts displayed 35% lower HA intensity per HA aggregate, when compared with control siblings (Fig. 9b), indicating that the ablation of *hapln1a*⁺ cells disrupted the linear structure of HA in the coronary growth area. Next, we examined HA signals in the regeneration area and found HA signals were closely associated with *deltaC:EGFP*⁺ cells (Fig. 9c). We then assessed the HA deposition within the regenerating area after depleting *hapln1a*⁺ cells and observed that 7 dpa injury sites of *hapln1a*⁺ cell-depleted animals displayed ~47% smaller puncta of HA, compared with wild-type siblings (Fig. 9d). Next, we assessed whether *serpine1* functions through regulating the HA structure during morphogenesis and regeneration, as previous reports indicated *serpine1* protects the ECM to maintain a matrix necessary for endothelial cells to migrate and form capillaries [86]. We examined HA deposition in the ventricle surface after treating juvenile *deltaC:EGFP* fish with Tiplaxtinin for 12 hours and continuously for 7 days and found that inhibitor treated hearts displayed a 64% lower HA intensity per HA aggregate, when compared with control siblings (Fig. 9e and f), revealing a disorganized HA linear structure within the coronary growth area. Lastly, we treated adult fish with Tiplaxtinin from 3 dpa for 12 hours and continuously for 4 days and observed 53% lower HA intensity per HA aggregate at 7 dpa, compared with vehicle controls (Fig. 9g and h). Our results indicate a mechanism for *hapln1a*⁺ cell function during morphogenesis and regeneration, in which HA is deposited by *hapln1a*⁺ cells and then organized by *serpine1*-expressing cells for coronary growth during vascularization and revascularization.”

3) More importantly, no experiments are carried out to show that there is anything special about the *hapln1a*⁺ cells beyond an association. The ablation of *tcf21*⁺ cells has been shown to impede coronary vessel growth and regeneration in previous publications. Here the authors show similar effects with ablation of *hapln1a*⁺ cells. However, the *hapln1a*⁺ cells comprise a significant part of the epicardium (33% in homeostasis, 50% in regenerating hearts??). The proper control is to ablate a similar proportion of *hapln1a*⁺ cells, to determine if the role in coronary growth is really specific to the *hapln1a*⁺ population. This is not a small ask, but is really essential for interpreting the results especially for a journal with impact of Nature Comm. At the very least, one could compare a partial 50% random ablation using *tcf21:NTR* with the full ablation of *hapln1a:NTR*. Does the latter show a much stronger phenotype?

A: The reviewer is insightful to point out this possibility. Currently, we have no method available to specifically ablate *hapln1a*⁺ cells. Therefore, we performed a partial (50%), random ablation of epicardial cells as suggested by the reviewer. After depleting around 50% of *tcf21*⁺ cells, we did not observe a significant difference in coronary vascularization during morphogenesis and revascularization after regeneration, compared with controls. These data indicate that ablating most *hapln1a*⁺ cells has a stronger phenotype on coronary growth, compared with spatial ablation of *tcf21*⁺ cells. We have included these new experiments on page 12, paragraph 1 and in Supplementary Fig. 5 of the revised manuscript:

“As *hapln1a*⁺ cells form around 30% of *tcf21*⁺ cells during morphogenesis (Fig. 1d) and around 50% during regeneration [23], we also examined the effect of randomly ablating

around 50% of *tcf21*⁺ cells (Supplementary Fig. 5a and b). With this method of *tcf21*⁺ cell ablation, there was no significant difference in coronary vascularization and revascularization when compared with control animals (Supplementary Fig. 5c-f), indicating that ablating *hapln1a*⁺ cells has a more severe effect on coronary growth.”

4) Details for inhibition of Serpine1 were missing. Is the concentration used equivalent to just what is needed to inhibit activity? Can inhibition of activity be measured? A complementary genetic approach would be highly desirable to exclude chemical off-targets.

A: We agree with the reviewer that a genetic approach is an excellent means to exclude possible chemical off-target effects and drug non-specific effects/efficiency. As the reviewer mentioned, genetic means of manipulating adult zebrafish genes are time-consuming and challenging/risky: 1) more than 1 year to develop a CRISPR/Cas9 *serpine1* mutant generation and several generation outcrosses; and 2) a *serpine1* mutant also does not guarantee that there will be no compensation effects. For these reasons, we chose a pharmacological inhibition approach to perform functional analysis of *serpine1* during heart regeneration. We chose the drug Tiplaxtinin as its *serpine1* inhibition activity has been previously measured and was utilized for the pharmacological inhibition of *serpine1* activity and published by different research groups (Gorlatova et al., JBC 2007; Daniel et al., PLoS One 2015; Kang et al., Oncotarget 2016; Munch et al., Development 2017; Lin et al., Cell Rep. 2020;; Yamagami et al., BBRC 2020). Natalia V. Gorlatova demonstrated the capacity for Tiplaxtinin to block *serpine1* activity, and IC₅₀ values for *serpine1* inactivation by Tiplaxtinin are from 9 to 12 μM. The inhibitory effect of 20 μM Tiplaxtinin for *serpine1*, the concentration which we used in this MS, is about 95%. As the means to examine possible downstream effectors of *serpine1*, like urokinase plasminogen activator (uPA) and tissue plasminogen activator (tPA) is currently still lacking in zebrafish, we are unable to examine the effect of *serpine1* inhibition. However, we performed experiments to exclude the possible toxic effects of Tiplaxtinin by examining coronary growth after removing the drug. We found that inhibited coronary growth restarted after removing the inhibitor. We have now included these data on pages 13-14, and in Supplementary Fig. 8 of the revised manuscript.

“...by treatment with its antagonist Tiplaxtinin, which blocks Serpine1 protease and has been used to inhibit *serpine1* activity in different systems, including during zebrafish heart regeneration [70, 73-77]. Previous studies measured and determined that the inhibitory effect of 20 μM of Tiplaxtinin on *serpine1* is around 95% [73, 78].”

“Moreover, these blocked coronary extensions can re-start after drug removal, indicating that the coronary growth restriction by *serpine1* inhibition does not occur by coronary cell death (Supplementary Fig. 8).”

Minor issues:

1) The term “shear structures” needs explanation as it is not a commonly used descriptor. Is there some evidence for causation of shear stresses?

A: We have updated the revised manuscript to include the phrase “linear structure” instead of “shear structure”.

2) What does it mean that vessel number is increased by hapln1a cell ablations? Are these different vessels or fragments of previous vessels?

A: As *hapln1a*⁺ cells also wrap around existing coronary vessels and behave as perivascular cells, we speculate that *hapln1a*⁺ cells play roles in vessel stabilization and the increased vessel numbers result from the fragmentation of newly formed vessels with *hapln1a*⁺ cell loss. We have included this information in page 11 of the revised manuscript:

“...while coronary vessel numbers in *hapln1a:NTR;deltaC:EGFP* animals were more than in *deltaC:EGFP* sibling controls (Fig. 4f). As *hapln1a*⁺ cells also envelop existing coronary vessels and behave as perivascular cells, these results indicate that coronary growth is not only blocked but coronary vessels also become unstable without *hapln1a*⁺ cells. This instability caused the fragmentation of existing vessels and resulted in the increased vessel numbers in the condition of *hapln1a*⁺ cell loss.”

Reviewer 3

In this study Sun et al identify a subset of epicardial cells that express Hapln1a which are associated with sprouting coronary vessels in the juvenile and regenerating zebrafish hearts, and use ablation experiments to show that Hapln1a is required for coronary growth in both scenarios. They further identify that serpine1, which encodes for a negative regulator of plasminogen activator, is expressed in hapln1a⁺ epicardial cells, is found at coronary tips, and is required for vascularisation both in coronary development and regeneration. Previous studies have identified that epicardial subtypes exist, that the epicardium supports coronary development, that hapln1a is expressed in the epicardium, that hapln1a is required for heart regeneration, and that serpine1 also promotes heart regeneration. However, little is understood about heterogeneity in the epicardium, how the epicardium supports coronary growth, and which epicardial cells are required. In this context, the study by Sun et al contributes new knowledge to the field of coronary development, and unifies and extends previous studies on the role of epicardium in vascularisation and revascularisation in development and injury. The experiments are well designed and executed, and the data is well presented. However, the depth of mechanistic analysis of the roles of hapln1a and serpine1 in vascularisation, particularly in terms of their functions and relationship, is a little unsatisfactory. What is the impact of Hapln1a and Serpine1 LOF on the ECM around the vasculature for example?

A: We appreciate the reviewer's comments and per the reviewer's suggestions we have performed additional experiments to further analyze the effect of *hapln1a*⁺ cells and *serpine1* on the ECM component hyaluronan (HA) for further understanding of the underlying mechanism. With our new analyses, we observed HA lining coronary vessel extensions and also preceding these vessels. Further, we discovered that depletion of *hapln1a*⁺ cells disrupted the HA organization and inhibiting *serpine1* activity led to a disorganized HA structure. Our new experiments indicate that HA deposition by *hapln1a*⁺ cells and then organization by *serpine1*-expressing cells is required for coronary growth during morphogenesis and regeneration. Although previous tissue engineering studies have demonstrated that HA-based materials are sufficient to stimulate angiogenesis and vascular growth and HA is required for zebrafish coronary re-vascularization (Brandes et al., JCI 1991; Peattie et al., Europe PMC 2004; Turner et al., Biomaterials 2004; Lepidi et al., Eur J Vasc Endovasc Surg 2006; Pardue et al., Organogenesis 2008; Burdick and Prestwich, Adv Mater 2011; Munch et al., Development 2017; Ghose et al., BMC Cancer 2018; Luo et al., Front Bioeng Biotechnol 2022), there is no direct observation of organized HA structures along and preceding coronary growth *in vivo*. Further, no reports exist regarding the regulation mechanism of HA by *hapln1a*⁺ cells and *serpine1*. We have updated the manuscript to include these new experiments on pages 14-16, and in Fig. 9 of the revised manuscript:

"To elucidate the mechanism by which *hapln1a*⁺ cells and *serpine1* activity impact coronary growth, we assessed the distribution of the *hapln1a*⁺ cell substrate, hyaluronic acid (HA). We recently reported that *hapln1a*⁺ epicardial cells regulate HA deposition to facilitate cardiomyocyte proliferation during heart morphogenesis and regeneration [48]. HA is required for coronary revascularization during zebrafish heart regeneration [79-81]. Previous studies also demonstrated that endothelial cells can attach and grow along HA fibers *in vitro*, and a scaffold in tissue-engineered vascular grafts formed from HA-based biomaterials can completely generate a new vascular tube. These studies indicate that

the administration of HA can enhance cell proliferation, adhesion, tubular sprout formation, and the migration of endothelial cells [79-85]. We speculate that HA is involved in guided coronary growth. To examine this possibility, we first assessed HA localization with coronary vessels on the ventricular surface in juvenile *deltaC:EGFP* animals. We detected strong association of linear HA signals with EGFP⁺ signals (Fig. 9a). Next, we examined the HA deposition after ablating *hapln1a*⁺ cells at 7 wpf. Juvenile *hapln1a:mCherry-NTR;deltaC:EGFP* animals and *deltaC:EGFP* siblings were treated with 10 mM Mtz and their hearts were collected for histological analysis. We assessed the size of HA aggregates by quantifying the area of each focus of HA and found that *hapln1a*⁺ cell-depleted hearts displayed 35% lower HA intensity per HA aggregate, when compared with control siblings (Fig. 9b), indicating that the ablation of *hapln1a*⁺ cells disrupted the linear structure of HA in the coronary growth area. Next, we examined HA signals in the regeneration area and found HA signals were closely associated with *deltaC:EGFP*⁺ cells (Fig. 9c). We then assessed the HA deposition within the regenerating area after depleting *hapln1a*⁺ cells and observed that 7 dpa injury sites of *hapln1a*⁺ cell-depleted animals displayed ~47% smaller puncta of HA, compared with wild-type siblings (Fig. 9d). Next, we assessed whether *serpine1* functions through regulating the HA structure during morphogenesis and regeneration, as previous reports indicated *serpine1* protects the ECM to maintain a matrix necessary for endothelial cells to migrate and form capillaries [86]. We examined HA deposition in the ventricle surface after treating juvenile *deltaC:EGFP* fish with Tiplaxtinin for 12 hours and continuously for 7 days and found that inhibitor treated hearts displayed a 64% lower HA intensity per HA aggregate, when compared with control siblings (Fig. 9e and f), revealing a disorganized HA linear structure within the coronary growth area. Lastly, we treated adult fish with Tiplaxtinin from 3 dpa for 12 hours and continuously for 4 days and observed 53% lower HA intensity per HA aggregate at 7 dpa, compared with vehicle controls (Fig. 9g and h). Our results indicate a mechanism for *hapln1a*⁺ cell function during morphogenesis and regeneration, in which HA is deposited by *hapln1a*⁺ cells and then organized by *serpine1*-expressing cells for coronary growth during vascularization and revascularization.”

How/why do these ECM modifications support vessel formation? And what is the relationship between them?

A: Previous tissue engineering studies have demonstrated that HA-based materials are sufficient to stimulate angiogenesis and vascular growth and HA is required for zebrafish coronary re-vascularization (Brandes et al., JCI 1991; Peattie et al., Europe PMC 2004; Turner et al., Biomaterials 2004; Lepidi et al., Eur J Vasc Endovasc Surg 2006; Pardue et al., Organogenesis 2008; Burdick and Prestwich, Adv Mater 2011; Munch et al., Development 2017; Ghose et al., BMC Cancer 2018; Luo et al., Front Bioeng Biotechnol 2022). However, there is no direct observation of organized HA structures along and preceding coronary growth *in vivo*. Further, no reports exist regarding the regulation mechanism of HA by *hapln1a*⁺ cells and *serpine1*. Our results indicate that HA lines the coronary vessel extensions and also precedes these vessels. These HA cables may function as scaffolds, which could prevent loss of ECM components during tissue remodeling, act as a template for matrix regeneration, and support interactions with other cell types (Evanko et al., J Histochem Cytochem 2009; Sun and Keller, Exp Eye Res 2015; Motte et al., JBC 1999; Wang and Hascall, JBC 2004; Jokela et al., Connect Tissue

Res. 2008). We have included these experiments on page 14-16, in Fig. 9 (Please see the response to the first Comment above); and on page 17, paragraph 1 of the revised manuscript:

“Our work indicates that *hapln1a*⁺ cells regulate HA organization during heart morphogenesis and regeneration. Although HA has previously been implicated in vessel growth stimulation and been applied for vessel tube formation during bioengineering [70, 79-85], no studies have reported that HA lines coronary vessels and paves the road for coronary growth during morphogenesis and endogenous regeneration. These *hapln1a*⁺ cell derived HA cables may function as scaffolds, which could prevent loss of ECM components during tissue remodeling, act as a template for matrix regeneration, and support interactions with other cell types [94-98].”

While further experiments would help to provide better insights into the specific role of the Hapln1a⁺ Serpine1⁺ epicardium in supporting coronary (re)development, the authors also do not really dive into these questions in the discussion, which is an omission. Taking steps to address the mechanism would significantly improve the impact of the manuscript.

A: We appreciate these comments and have now performed additional experiments to further explore the underlying mechanisms of *hapln1a*⁺ cells and *serpine1* on coronary growth. We uncovered that *hapln1a*⁺ cells/*serpine1* control coronary growth by regulating the deposition/organization of HA, which was previously indicated to be sufficient to stimulate vessel growth by bioengineering (Brandes et al., JCI 1991; Peattie et al., Europe PMC 2004; Turner et al., Biomaterials 2004; Lepidi et al., Eur J Vasc Endovasc Surg 2006; Pardue et al., Organogenesis 2008; Burdick and Prestwich, Adv Mater 2011; Munch et al., Development 2017; Ghose et al., BMC Cancer 2018; Luo et al., Front Bioeng Biotechnol 2022). We have included these results and information on pages 14-16 and Fig. 9 of the revised manuscript. We also included further discussion of *hapln1a*⁺ cells and *serpine1* function in coronary sprouting/growth on page 17 of the revised manuscript. Please see the responses to the first and second Comments above.

Major comments:

1. What is the impact of Hapln1a and Serpine1 LOF on the ECM around the vasculature for example?

A: We have now performed HA analyses after depleting *hapln1a*⁺ cells and inhibiting *serpine1* activity. We observed that the HA structure is disorganized around the vessels with *hapln1a*⁺ cell or *serpine1* activity disruption, when compared with controls. We included these results on pages 15-16 and Fig. 9 in the revised manuscript.

“...To examine this possibility, we first assessed HA localization with coronary vessels on the ventricular surface in juvenile *deltaC:EGFP* animals. We detected strong association of linear HA signals with EGFP⁺ signals (Fig. 9a). Next, we examined the HA deposition after ablating *hapln1a*⁺ cells at 7 wpf. Juvenile *hapln1a:mCherry-NTR;deltaC:EGFP* animals and *deltaC:EGFP* siblings were treated with 10 mM Mtz and their hearts were collected for histological analysis. We assessed the size of HA aggregates by quantifying the area of each focus of HA and found that *hapln1a*⁺ cell-depleted hearts displayed 35% lower HA intensity per HA aggregate, when compared with control siblings (Fig. 9b), indicating that the ablation of *hapln1a*⁺ cells disrupted the linear structure of HA in the coronary growth area. Next, we examined HA signals in the regeneration area and found

HA signals were closely associated with *deltaC:EGFP*⁺ cells (Fig. 9c). We then assessed the HA deposition within the regenerating area after depleting *hapln1a*⁺ cells and observed that 7 dpa injury sites of *hapln1a*⁺ cell-depleted animals displayed ~47% smaller puncta of HA, compared with wild-type siblings (Fig. 9d). Next, we assessed whether *serpine1* functions through regulating the HA structure during morphogenesis and regeneration, as previous report indicated *serpine1* protects the ECM to maintain a matrix necessary for endothelial cells to migrate and form capillaries [86]. We examined HA deposition in the ventricle surface after treating juvenile *deltaC:EGFP* fish with Tiplaxtinin for 12 hours and continuously for 7 days and found that inhibitor treated hearts displayed a 64% lower HA intensity per HA aggregate, when compared with control siblings (Fig. 9e and f), revealing a disorganized HA linear structure within the coronary growth area. Lastly, we treated adult fish with Tiplaxtinin from 3 dpa for 12 hours and continuously for 4 days and observed 53% lower HA intensity per HA aggregate at 7 dpa, compared with vehicle controls (Fig. 9g and h).”

2. How/why do these ECM modifications support vessel formation? And what is the relationship between them?

A: Previous tissue engineering studies have demonstrated that HA-based materials are sufficient to stimulate angiogenesis and vascular growth, and the administration of HA can enhance cell proliferation, adhesion, tubular sprout formation, and the migration of endothelial cells (Brandes et al., JCI 1991; Peattie et al., Europe PMC 2004; Turner et al., Biomaterials 2004; Lepidi et al., Eur J Vasc Endovasc Surg 2006; Pardue et al., Organogenesis 2008; Burdick and Prestwich, Adv Mater 2011; Munch et al., Development 2017; Ghose et al., BMC Cancer 2018; Luo et al., Front Bioeng Biotechnol 2022). However, there are no reports on the organized HA structures along and preceding coronary growth and the regulation mechanism of HA *in vivo*. Our discoveries indicate that *hapln1a*⁺ cells/*serpine1* control coronary growth by regulating the deposition/organization of HA. We included these results on pages 15-16 in the revised manuscript:

“HA has been indicated to be required for coronary revascularization during zebrafish heart regeneration [79-81]. Previous studies also demonstrated that endothelial cells can attach and grow along HA fibers *in vitro*, and a scaffold in tissue-engineered vascular grafts formed with HA-based biomaterials can completely generate a newly formed vascular tube. These studies indicate the administration of HA can enhance cell proliferation, adhesion, tubular sprout formation, and the migration of endothelial cells [79-85].”

“Our results indicate a mechanism for *hapln1a*⁺ cell function during morphogenesis and regeneration, in which HA is deposited by *hapln1a*⁺ cells and then organized by *serpine1*-expressing cells for coronary growth during vascularization and revascularization.”

3. Further experiments would help to provide better insights into the specific role of the *Hapln1a*⁺ *Serpine1*⁺ epicardium in supporting coronary (re)development, the authors also do not really dive into these questions in the discussion, which is an omission.

A: Our previous study demonstrated that *hapln1a*⁺ cells are required for the synthesis and organization of the ECM component hyaluronic acid (HA) (Sun et al., Circulation 2022). Additionally, other reports revealed that *Serpine1* protects the ECM to maintain a

matrix necessary for endothelial cells to migrate and form capillaries (Ismail et al., Int J Mol Sci. 2022). To address this question, with our new analyses, we observed HA lining coronary vessel extensions and also preceding these vessels. Further, we discovered that depletion of *hapln1a*⁺ cells disrupted the HA organization and inhibiting *serpine1* activity led to a disorganized HA structure. Our new experiments indicate that HA deposition by *hapln1a*⁺ cells and then organization by *serpine1*-expressing cells is required for coronary growth during morphogenesis and regeneration. These results indicate that *hapln1a*⁺ cells and *serpine1* activity are involved in the establishment of an organized HA structure that is required for continuous coronary growth. We have included in the results on pages 14-16, Fig. 9, and in the discussion on page 17:

“Our work indicates that *hapln1a*⁺ cells regulate HA organization during heart morphogenesis and regeneration. Although HA has previously been implicated in vessel growth stimulation and been applied for vessel tube formation during bioengineering [70, 79-85], no studies have reported that HA lines coronary vessels and paves the road for coronary growth during morphogenesis and natural regeneration. These *hapln1a*⁺ cell derived HA cables may function as scaffolds, which could prevent loss of ECM components during tissue remodeling, act as a template for matrix regeneration, and support interactions with other cell types [94-98].”

4. Relating to the comment above, the authors do not investigate changes in ECM in either the Hapln1a or Serpine1 models, or discuss this mechanism of action of each gene. The authors could assess for example HA presence/organisation (as in their previous Circulation Hapln1a epicardium paper) around the vessels in the juvenile stage in wild type and upon ablation of Hapln1a⁺ cells or Tiplaxtinin treatment.

A: We appreciate these comments and have performed HA analyses. Please see our response to the first Comment above.

5. In Line 339 – the authors also suggest that *hapln1a* regulates coronary vessel development through *Serpine1* and that they will test this – of course this will not be happening at a transcriptional level, and since the authors also don't look in detail at the mechanisms of action of either protein, or the impact on the extracellular environment (showing only that they both promote coronary development), the experiments that they perform subsequently don't really address this hypothesis.

A: We have examined our language in the manuscript based on this suggestion. We feel it is accurate to state that:

“*hapln1a*⁺ cells precede coronary growth extensions and express *serpine1*, and these *hapln1a*⁺ cells and the *serpine1* gene are required for an ECM environment through HA organization.”

6. It appears that not all vessel tips are associated with Hapln1a (or even Serpine1) for example the 6 hour timepoint shown in Figure 2 – there appear to be sprouts forming without any Hapln1a. Related to this, some revascularization still occurs in the *hapln1a* KO. Does this mean Hapln1a is only required by a subset of branching vessels? To investigate this, the authors should quantify the number of tips with/without Hapln1a and Serpine1, and comment on this possible heterogeneity.

A: The reviewer is insightful to point out this possibility. To be accurate, we have changed the word “coronary tip” to “coronary sprout”. We agree with the reviewer that coronary sprouting can form without any *hapln1a*⁺ cells. Our analyses revealed that there is a small percentage (around 18%) of coronary extensions that occur without any *hapln1a*⁺ cells during heart morphogenesis. Although we also observed the withdrawal of the extended coronary vessels without *hapln1a*⁺ cells, we cannot exclude other mechanisms that provide guidance cues that may affect coronary growth during vascularization and revascularization. There are other possibilities such as: macrophages affect vessel growth, as we also observed macrophages attached to vessels in some areas. We have updated the revised manuscript to include discussion of coronary vessel tips without *hapln1a*⁺ cells and possible other guidance cues they depend on for growth on page 18, paragraph 1:

“In this study, we discovered that *hapln1a*⁺ cells and *serpine1* activity are responsible for major coronary growth in zebrafish: *hapln1a*⁺ cells deposit HA and *serpine1* regulates the HA organization to form a linear structure, which is required for coronary extension and continuous growth. Further, our analyses also revealed that there is a small percentage (around 18%) of coronary growth extensions that occur without any *hapln1a*⁺ cells during heart morphogenesis. Although we also observed the withdrawal of the extended coronary vessels without *hapln1a*⁺ cells, we cannot exclude other mechanisms that provide guidance cues that may affect coronary growth during vascularization and revascularization.”

7. The authors state that the location of *hapln1a* cells indicates they function as perivascular cells (Line 221) – are there perivascular markers in the *hapln1a*⁺ clusters? If so, which cluster?

A: We examined the scRNA-seq data of *tcf21*⁺ cells, and found that the perivascular marker, *pdgfrb*, is expressed in the *tcf21*⁺ cell clusters. This information is now included on page 7, paragraph 2, and Supplementary Fig. 4 of the revised manuscript:

“We then analyzed the scRNA-seq data for perivascular cell markers and observed *pdgfrβ* expression in clusters 1 and 6 [50] (Supplementary Fig. 4).”

8. The authors also identify 3 different kinds of *hapln1a*⁺ cell – 2 of which behave more like perivascular cells, surrounding the vessels, and 1 of which (the most prevalent kind, which ‘precedes’ the growing sprout and is *Serpine1* + in juveniles) exhibits behaviour unexpected of a perivascular cell. Do the authors think these cell types are from different *Hapln1*⁺ clusters, or are they all the same subtype? In the single cell seq data presented in figure S2 (*Hapln1*⁺ sorted), *serpine1* is found in a relatively small proportion of those cells – do these form a distinct cluster separate from other *Hapln1a*⁺ cells? Does clustering of the *Hapln1a*⁺ sc-RNA-seq data give more insights into this population?

A: We appreciate these thoughtful comments and indeed performed further analyses with our scRNA-seq data. Unfortunately, we couldn’t separate *serpine1*-expressing cells from other *hapln1a*⁺ cells. We speculate that this occurred as *hapln1a*-expressing cells are a subtype of epicardial cells and the *hapln1a*⁺ cells are likely a relatively homogenous cell population, with different transitional states leading to difficulties with cluster separation. We are in the process of generating BAC and knock-in *serpine1* fluorescence reporters and will isolate these cells for deep sequencing to perform comparisons between

hapln1a⁺/serpine1⁻ and *hapln1a⁺/serpine1⁺* cells during morphogenesis and regeneration in future studies.

9. In Fig S2, *serpine1* is expressed at very low levels in *Hapln1a⁺* cells of juvenile hearts compared to regenerating hearts – the authors should comment on this difference.

A: It was previously reported that *serpine1* is also expressed in endocardial cells in the inner injured area during heart regeneration (Munch et al., Development 2017). As coronary growth occurs in the ventricular wall and revascularization occurs in the lateral injury area at early stages of regeneration, we speculate that the inner endocardium-derived *serpine1* is not involved in coronary growth guidance. We have included this information on page 14, paragraph 1:

“Similar to a previous report that the endocardium expresses *serpine1* after heart injury [70], we also detected *serpine1* signals in the inner injury area. However, we speculate that these *serpine1*-expressing cells are not directly correlated with coronary extension as coronary vessels mainly extended from the lateral area of the ventricular wall to the middle of the injury site. Together, our results indicate that *hapln1a⁺* cell-derived *serpine1* controls coronary growth during heart morphogenesis and regeneration.”

and page 18, paragraph 2 of the revised manuscript:

“Further, we noticed that the percentage of *serpine1* expression in *hapln1a⁺* cells is different during heart morphogenesis and regeneration. We speculate that this difference arose because coronary vascularization occurs for around two months (from 5 wpf -12 wpf) and builds the coronary vasculature from very few coronary cells, while coronary revascularization occurs within 2-3 weeks and generates new vessels from existing vessels. Another possibility is that the endocardium functions through *serpine1* to regulate myocardial regeneration while such endocardium-derived *serpine1* doesn't exist during heart morphogenesis. We speculate that these differences led us to detect low expression levels of *serpine1* in morphogenesis hearts in comparison with *serpine1* expression in the injury site.”

Related to this, the phenotype in the *Serpine1*-inhibited hearts at 7wpf (Fig. 7C) is very profound given the small number of cells that express *Serpine1* (Fig S2). In the explanted hearts (Fig 7A), there even appears to be a reduction in vessels – is this the impact not of defective growth, but rather death/regression of the existing vascular network? The authors should comment on these issues.

A: Our previous published work indicated that coronary growth occurs in juvenile fish hearts (Sun et al., Dev Bio. 2022), which indicates that there are newly formed coronary vessels that exist in the ventricular wall. These new vessels (immature vessels) need to be stabilized by ECM and perivascular cells. We speculate that although *serpine1* is expressed in a small number of *hapln1a⁺* cells, these *serpine1⁺* cells are important for the formation of correct HA structures that stabilize newly formed vessels and for continuous growth of extending coronary sprouts. If *serpine1* activity is inhibited, newly formed coronary vessels will become unstable and growing coronary vessels will withdraw. Further, we performed *ex vivo* culture of juvenile hearts with a *serpine1* inhibitor and observed blocked coronary growth, and then detected coronary growth after removal of this drug. This experiment indicates that the regression is not from death of existing

vessels. This information has been included in our revised manuscript on page 14 and Supplementary Fig. 8:

“Moreover, these blocked coronary extensions can re-start after drug removal, indicating that the coronary growth restriction by *serpine1* inhibition does not occur by coronary cell death (Supplementary Fig. 8).”

Minor comments:

The authors state there is significant enrichment of Hapln1a in the angiogenic epicardial subclusters of 1 and 3 (Line 208). In Figure 1 it is clear there is also significant expression in clusters 4 and 5, and so this statement is a little misleading (arguably, cluster 4 actually has more hapln1a+ expressing cells than cluster 3).

A: We agree with the reviewer and have now clarified *hapln1a* expression in the revised manuscript. The Violin plot analysis of *hapln1a* indicated high expression of *hapln1a* in cluster 1, slight expression in cluster 3, 4, 5, and little expression in clusters 2 and 6. We have revised the description on page 7, paragraph 1:

“As cluster 1 is the most heavily represented of these 3 cell states, we first focused on cluster 1 and observed high expression of *hapln1a* in this cluster [23], with slight expression in clusters 3, 4 and 5 (Fig. 1d and Supplementary Fig. 3).”

Did the authors confirm that the Tiplaxtinin treatment is blocking *serpine1* activity/function? For example by assaying conversion of plasminogen to plasmin.

A: We chose the drug Tiplaxtinin as its *serpine1* inhibition activity has been previously measured and was utilized for the pharmacological inhibition of *serpine1* activity and published by different research groups (Gorlatova et al., JBC 2007; Daniel et al., PLoS One 2015; Kang et al., Oncotarget 2016; Munch et al., Development 2017; Lin et al., Cell Rep. 2020; Yamagami et al., BBRC 2020). Natalia V. Gorlatova demonstrated the capacity for Tiplaxtinin to block *serpine1* activity, and IC₅₀ values for *serpine1* inactivation by Tiplaxtinin are from 9 to 12 μM. The inhibitory effect of 20 μM Tiplaxtinin for *serpine1*, the concentration which we used in this MS, is about 95%. As the means to examine possible downstream effectors of *serpine1*, like urokinase plasminogen activator (uPA) and tissue plasminogen activator (tPA) is currently still lacking in zebrafish, we are unable to examine the effect of *serpine1* inhibition. However, we performed experiments to exclude possible toxic effects of Tiplaxtinin by examining coronary growth after removing the drug. We found that inhibited coronary growth restarted after removing the inhibitor. We have now included these data on pages 13-14, and in Supplementary Fig. 8 of the revised manuscript.

“...by treatment with its antagonist Tiplaxtinin, which blocks *Serpine1* protease and has been used to inhibit *serpine1* activity in different systems, including during zebrafish heart regeneration [70, 73-77]. Previous studies measured and determined that the inhibitory effect of 20 μM of Tiplaxtinin on *serpine1* is around 95% [73, 78].”

“Moreover, these blocked coronary extensions can re-start after drug removal, indicating that the coronary growth restriction by *serpine1* inhibition does not occur by coronary cell death (Supplementary Fig. 8).”

Figure 1 – there is a lack of consistency in the colour coding used in the heatmaps -

sometimes blue represents the highest value, sometimes the lowest – this could be consistent within the figure to simplify.

A: We have changed the color to be consistent as recommended.

Figure 4 – The authors show that the vessel density and junction number (graphs D and E) are reduced, which is in line with the observation that vascular growth is less in the *hapln1a* ablation model. However, vessel number per area is increased (F), which seems contradictory to the previous quantifications. Can the authors explain the difference between these two quantifications?

A: As *hapln1a*⁺ cells also wrap around existing coronary vessels and behave as perivascular cells, we speculate that *hapln1a*⁺ cells play roles in vessel stabilization and the increased vessel numbers result from the fragmentation of newly formed vessels with *hapln1a*⁺ cell loss. We have included these information in page 11 of the revised manuscript:

“...while coronary vessel numbers in *hapln1a:NTR;deltaC:EGFP* animals were more than in *deltaC:EGFP* sibling controls (Fig. 4f). As *hapln1a*⁺ cells also envelop existing coronary vessels and behave as perivascular cells, these results indicate that coronary growth is not only blocked but coronary vessels also become unstable without *hapln1a*⁺ cells. This instability caused the fragmentation of existing vessels and resulted in the increased vessel numbers in the condition of *hapln1a*⁺ cell loss.”

Figure 4 and 5 – the graphs are distorted along the x axis – aspect ratio should be maintained.

A. We have corrected these graphs as recommended.

Figure 6 – The *serpine1* expression shown in the images doesn't seem to colocalise with *Hapln1a* at terminal branches (rather it is localised more centrally within the *Hapln1a* network) – is *Serpine1* expressed broadly within the *Hapln1a* network as well as around tip cells?

A: We observed that *hapln1a*⁺ cells (which deposit HA) and HA deposition can precede coronary growth sprouts over a long distance. The *serpine1* is expressed in *hapln1a*⁺ cells close to the coronary sprouts and involved in organizing the HA structure, leading to the appearance of *serpine1* localizing more centrally within *hapln1a*⁺ cell linear structures.

REVIEWER COMMENTS

Reviewer #1 (Remarks to the Author):

The authors have done a very good job of addressing all issues I had raised. Genetic confirmation of the role of serpin1 is still missing, but I appreciate that this is very challenging, and I am fine with the additional control they did to show that the drug seems to not be overly toxic.

I thus can now recommend publication in Nat. Comm.

Reviewer #2 (Remarks to the Author):

This is a significantly improved manuscript following quite extensive revisions. Many conflicting or unclear statements were corrected, modified, or clarified. A major improvement includes incorporation of data on HA deposition as part of the mechanisms driving coronary growth. My previous concerns were largely addressed and I have no further major concerns.

Reviewer #3 (Remarks to the Author):

In this revised manuscript, Sun et al have satisfactorily addressed the majority of my comments. In particular they have expanded/dived a little deeper into their sc-RNA seq data, and discussed more fully the regeneration data they present.

The only major area where the manuscript is still weaker is mechanism. The authors have included an extra set of experiments where they use an HA-binding antibody to assess HA deposition/organisation around sprouting coronaries. They show that HA is organised in a linear fashion around vessels in wild type hearts, and show that linear HA deposition is disrupted in both Hapln1a KO and Serpine KO. The authors then conclude that HA is deposited by hapln1a+ cells and then organized by serpine1-expressing cells for coronary growth during vascularization and revascularization.

I have a few questions around the interpretation of this data, that are important for developing the mechanistic insights this paper endeavours to deliver.

1) Hapln1a KO appears to lead to large reduction of HA in both models, whereas in the juvenile Serpine KO the impact is much less than either the Hapln1a KO or Serpine regeneration model (linear structures appear better preserved, for example). Why this difference?

2) The authors seem to be suggesting that hapln1a+ cells deposit the HA, and Serpine organises it (stated in the results section and again in the discussion). To support this theory, do they find that HA synthesising genes are expressed specifically in their hapln1a+ clusters from their sc-RNA seq data?

3) Related to this, Hapln1a itself is an HA-binding protein – so why do they think that Hapln1a cells function to produce the HA, but only Serpine is organising it? Could Hapln1a also not be organising the HA? The authors do not really seem to consider this role of Hapln1a, but this would be a relatively logical function given the nature of Hapln1a.

4) High and low molecular weight HA can have different biological functions. Is the reduction in HA in the Hapln1a and Serpine1 models a failure to produce HA, failure to organise HA, failure to protect HA from degradation? If the HA is degraded, then smaller bioactive fragments may be produced, which could influence tissue differently (indeed, cleaved HA fragments have already been shown to be important for vessel formation in zebrafish embryos). The implications of loss of Hapln1a and Serpine could be widespread depending on the wild type requirements and mechanisms of HA - what form the HA is in, how it is turned over and what kind of signalling/scaffolding roles it is playing.

A couple of minor comments:

1) Details on HA quantification I think are missing from the methods section

2) There is a halpn1a/hapln1a typo on page 18

Below is a point-by-point response to the reviewer's comments.

Reviewer 3

In this revised manuscript, Sun et al have satisfactorily addressed the majority of my comments. In particular they have expanded/dived a little deeper into their sc-RNA seq data, and discussed more fully the regeneration data they present.

The only major area where the manuscript is still weaker is mechanism. The authors have included an extra set of experiments where they use an HA-binding antibody to assess HA deposition/organisation around sprouting coronaries. They show that HA is organised in a linear fashion around vessels in wild type hearts, and show that linear HA deposition is disrupted in both Hapln1a KO and Serpine KO. The authors then conclude that HA is deposited by hapln1a+ cells and then organized by serpine1-expressing cells for coronary growth during vascularization and revascularization.

I have a few questions around the interpretation of this data, that are important for developing the mechanistic insights this paper endeavours to deliver.

1) Hapln1a KO appears to lead to large reduction of HA in both models, whereas in the juvenile Serpine KO the impact is much less than either the Hapln1a KO or Serpine regeneration model (linear structures appear better preserved, for example). Why this difference?

A: We agree with the reviewer that a stronger disruption effect on the HA structure occurred under conditions of depleting *hapln1a*⁺ cells in juvenile and regenerating hearts, and after blocking *serpine1* in regenerating hearts, when compared with *serpine1* blockage in juvenile hearts. Our previous report indicated that the HA synthesis enzyme *has1* is expressed in cluster 1, and *hapln1a*⁺ cell loss resulted in reduced HA production and disruption of the HA structure (Sun et al., Circulation 2022). As *serpine1* is expressed in *hapln1a*⁺ cells near the coronary extension area, we speculate that blocking *serpine1* leads to more local effects on the organization of HA. Further, although the heart regeneration process always recapitulates the developmental process, coronary vascularization in juvenile fish takes around 8 weeks (from 5 to 12 wpf) while revascularization occurs within 2-3 weeks. This difference may cause variations in the speed of ECM deposition and coronary growth during development and regeneration. Accordingly, the blockage of *serpine1* during development may cause less disruption on the structure of HA in juvenile hearts when compared with regenerating hearts. We have now included these discussions on page 19 of the revised manuscript.

“A lower disruption effect on the HA structure was observed after blocking *serpine1* in juvenile hearts when compared with other conditions such as depleting *hapln1a*⁺ cells in juvenile and regenerating hearts and blocking *serpine1* in regenerating hearts. One reason for these variations is that *hapln1a*⁺ cell loss not only leads to small aggregates of HA but also less HA production (Sun et al., Circulation 2022), while *serpine1* plays an organizational role in the local coronary extension area. Further, although the heart regeneration process always recapitulates the developmental process, coronary vascularization in juvenile fish takes around 2 months (from 5 wpf to 12 wpf), while revascularization occurs within 2-3 weeks. This temporal difference may cause different

speeds of ECM deposition and coronary growth during development and regeneration. Accordingly, blocking *serpine1* during development may cause a lower disruption effect on HA structure in juvenile hearts when compared with regenerating hearts.”

2) The authors seem to be suggesting that hapln1a+ cells deposit the HA, and Serpine organises it (stated in the results section and again in the discussion). To support this theory, do they find that HA synthesising genes are expressed specifically in their hapln1a+ clusters from their sc-RNA seq data?

A: We appreciate the reviewer’s comments. In our previous report, we identified expression of the HA synthesis enzyme *has1* in epicardial cluster 1, which largely expresses *hapln1*. We speculate that a sophisticated regulation system exists in epicardial cells to control ECM components like HA, to facilitate the microenvironment for coronary stabilization/growth/extension: the epicardial cluster (*has1⁺/hapln1⁺*) deposits HA and stabilizes HA binding with other ECM proteins like proteoglycan, and then local *serpine1* further organizes the linear structure to support coronary extension. We have included these discussions on page 18 of revised manuscript:

“Our previous report indicated that the epicardial cluster 1 not only deposits HA through *has1* expression but also requires *hapln1* to stabilize HA structures (Sun et al., Circulation 2022). Our current results indicated that *serpine1* is involved in this process by locally organizing linear HA structures to guide coronary extension.”

Further, we provided the reviewer our results on the expression of *has1* in juvenile, maintenance, and regenerating hearts, in the reviewer only supplemental materials. We found that *has1* is expressed in *hapln1a⁺* cells in juvenile and regenerating hearts. Our results suggested that the regulation of *has1* expression is coordinated with heart development and regeneration. Because we (Wang lab) are preparing another manuscript focusing on identifying *has1* expression regulation and the behaviors of *has1⁺/hapln1⁺* cells and *hapln1⁺/serpine1⁺* cells during development and regeneration, we ask that these results remain for review purposes only.

3) Related to this, Hapln1a itself is an HA-binding protein – so why do they think that Hapln1a cells function to produce the HA, but only Serpine is organising it? Could Hapln1a also not be organising the HA? The authors do not really seem to consider this role of Hapln1a, but this would be a relatively logical function given the nature of Hapln1a.

A: We apologize for this omission and have provided an explanation in the revised manuscript and here. In our previous report, we indicated that HA synthesis enzyme *has1* and *hapln1* are both expressed in epicardial cluster 1, and *hapln1* mutation resulted in a disrupted HA organization (Sun et al., Circulation 2022). As *hapln1* fluorescence reporters revealed that *hapln1* expression is distributed in the cells wrapping and in advance of coronary vessels, we speculate that the hapln1 protein plays a more general role to stabilize the binding of HA with other ECM proteins like proteoglycan, as pointed out by the reviewer. However, as *hapln1* is not expressed locally in the area of coronary extension, we don’t think it plays a major role in guidance. For this reason, we focused on *serpine1*, which functions in regulating neuron migration and cancer invasion, and is

locally expressed in the coronary extension area. We have included these discussions on page 18 of the revised manuscript:

“As *hapln1* expression is distributed in the cells wrapping and in advance of coronary vessels, we speculate that the *hapln1* protein plays a general role in organizing HA structure by stabilizing the binding of HA with other ECM proteins like proteoglycan, but it likely does not play a major role in the guidance of coronary growth as it is not expressed locally in the area of coronary extension.”

4) High and low molecular weight HA can have different biological functions. Is the reduction in HA in the *Hapln1a* and *Serpine1* models a failure to produce HA, failure to organise HA, failure to protect HA from degradation? If the HA is degraded, then smaller bioactive fragments may be produced, which could influence tissue differently (indeed, cleaved HA fragments have already been shown to be important for vessel formation in zebrafish embryos). The implications of loss of *Hapln1a* and *Serpine1* could be widespread depending on the wild type requirements and mechanisms of HA - what form the HA is in, how it is turned over and what kind of signalling/scaffolding roles it is playing.

A: The reviewer is insightful to point out this possibility. In our previous report (Sun et al., *Circulation* 2022), we found that *hapln1a*⁺ cell loss resulted in less HA production and disorganization of HA structures, while *hapln1* mutation causes disruption of HA structure. Our previous and current data suggested that the HA synthesis enzyme (*has1*), *hapln1*, and *serpine1* genes are players for HA regulation, but have different roles in deposition and organization. We have considered the possible existence and effect of high and low molecular weight HA in juvenile and regenerating hearts, as the reviewer mentioned and suggested. As the methods to examine molecular weight of HA is limited in juvenile and regenerating hearts, we currently cannot assess whether HA degradation plays roles in coronary growth. However, interestingly, we have observed the existence of *hapln1a*⁺ cells during the embryonic stage and their correlation with trunk vessels, as we observed that *hapln1a*⁺ cells line trunk vessels laterally. At this time, we prefer for these images to be limited to review only and are provided in the reviewer only supplemental materials, as we are interested in studying the possible role of *hapln1*⁺ cells in vessel growth during embryogenesis, which is an appropriate system to explore *hapln1*⁺ cell interaction/coordination with the players in regulating HA synthesis, organization, and degradation/cleavage as the reviewer suggested.

A couple of minor comments:

1) Details on HA quantification I think are missing from the methods section.

A: We have included the updated details of HA quantifications in the Methods section on pages 25 in the revised manuscript.

2) There is a *halpn1a*/*hapln1a* typo on page 18.

A. We have now corrected the manuscript.

REVIEWERS' COMMENTS

Reviewer #3 (Remarks to the Author):

Many thanks to the authors for their consideration of points raised in my previous review. I am happy with the extra additions to the manuscript text, and have no further comments.